# THE BENEFITS OF BEING CATEGORICAL DISTRIBUTIONAL: UNCERTAINTY-AWARE REGULARIZED EXPLORATION IN REINFORCEMENT LEARNING

## ABSTRACT

Despite the remarkable empirical performance of distributional reinforcement learning (RL), its theoretical advantages over classical RL are still not fully understood. Starting with Categorical Distributional RL (CDRL), we propose that the potential superiority of distributional RL can be attributed to a derived distribution-matching regularization by applying a return density function decomposition technique. This less-studied regularization in the distributional RL context aims to capture additional knowledge of return distribution beyond only its expectation, contributing to an augmented reward signal in policy optimization. In contrast to the standard entropy regularization in MaxEnt RL, which explicitly encourages exploration by promoting diverse actions, the regularization derived from CDRL implicitly updates policies to align the learned policy with environmental uncertainty. Finally, extensive experiments substantiate the significance of this uncertainty-aware regularization derived from distributional RL on the empirical benefits over classical RL. Our study offers a new perspective from the exploration to explain the benefits of adopting distributional learning in RL.

## 1 INTRODUCTION

The fundamental characteristics of classical reinforcement learning (RL) algorithms, such as Q-learning (Sutton & Barto, 2018; Watkins & Dayan, 1992), relies on estimating the expectation of discounted cumulative rewards that an agent observes while interacting with the environment. In contrast to the expectation-based RL, a novel branch of algorithms, termed *distributional RL*, seeks to estimate the entire distribution of total returns and has achieved state-of-the-art performance across a diverse array of environments (Bellemare et al., 2017a; Dabney et al., 2018b;a; Yang et al., 2019; Zhou et al., 2020; Nguyen et al., 2020; Wenliang et al., 2024; Sun et al., 2024b). Meanwhile, distributional RL inherits enhanced capabilities in areas, such as risk-sensitive control (Dabney et al., 2018a; Lim & Malik, 2022; Chen et al., 2024), offline learning (Wu et al., 2023; Ma et al., 2021), policy exploration (Cho et al., 2023; Mavrin et al., 2019; Rowland et al., 2019; Sun et al., 2024b), robustness (Sun et al., 2023; Sui et al., 2023), optimization (Sun et al., 2024a; Rowland et al., 2023; Kuang et al., 2023), and statistical inference (Zhang et al., 2023).

**Motivation: Interpreting the Benefits of Being (Categorical) Distributional in RL.** Despite the impressive empirical success of various distributional RL algorithms, our comprehension of their advantages in RL, especially within the general function approximation framework and practical implementations, remains incomplete. Early work (Lyle et al., 2019) demonstrated that in many realizations of tabular and linear approximation settings, distributional RL behaves similarly to classic RL, suggesting that its benefits are mainly realized in the non-linear approximation setting. Although their findings offer profound insights, their analysis, based on a coupled update method, overlooks several factors, such as the optimization effect under various losses. The statistical benefits of quantile temporal difference (QTD), employed in quantile distributional RL algorithms like QR-DQN (Dabney et al., 2018b), were highlighted in (Rowland et al., 2024; 2023), which posited that the robust estimation of QTD fosters the benefits in stochastic environments. The foundational theoretical aspects of CDRL were first discussed in (Rowland et al., 2018); however, the empirical superiority of adopting categorical distributional remains under-explored. Furthermore, recent studies (Wang et al., 2023; 2024) elucidate the benefits of distributional RL by introducing the novel

small-loss and second-order PAC bounds, demonstrating enhanced sample efficiency in specific cases, such as those with small achievable costs. Yet, their findings are not directly based on typical distributional RL algorithms commonly used in practice, such as C51 (Bellemare et al., 2017a) or QR-DQN. Therefore, it is imperative to close this gap between the theoretical explanation and practical deployment for distributional RL algorithms.

**Contributions.** In this paper, we interpret the potential advantages of distributional learning in RL over classical RL, specifically focusing on CDRL, the pioneering family within distributional RL. We examine these benefits through the lens of regularized exploration effect, offering a distinct perspective relative to existing literature. Our investigation begins with the decomposition of CDRL's objective function into an expectation-based term and a distribution-matching regularization, facilitated by our proposed return density decomposition technique. This regularization acts as an augmented reward in the actor-critic framework, encouraging policies to explore states and actions whose current return distribution estimates lag far behind the target ones, determined by environmental uncertainty. This derived regularization from the objective function of distributional learning promotes an uncertainty-aware exploration effect, diverging from the commonly used exploration for diverse actions in MaxEnt RL (Williams & Peng, 1991; Haarnoja et al., 2018a;b). Additionally, we propose a theoretically grounded algorithm called *Distribution-Entropy-Regularized Actor Critic*, interpolating between expectation-based and distributional RL. Empirical evidence underscores the pivotal role of the uncertainty-aware entropy regularization in CDRL's empirical success over expectation-based RL on both Atari games and MuJoCo environments. We further elucidate the distinct roles that the uncertainty-aware entropy in distributional RL and the explicit vanilla entropy in MaxEnt RL play by exploring their mutual impacts on learning performance. This opens new avenues for future research in this domain. Our contributions are summarized as follows:

- We propose a return density decomposition technique to decompose the objective function in CDRL. We argue that the derived regularization can promote uncertainty-aware exploration, which interprets the benefits of adopting distributional learning in RL.

- Within the actor-critic framework, we compare the cross-entropy-based uncertainty-aware regularization from distributional RL and vanilla entropy regularization in MaxEnt RL. A byproduct interpretable algorithm is further introduced, interpolating between classical and distributional RL.

- Empirically, we verify the uncertainty-aware regularization effect on the performance advantage of distributional RL and explore the mutual impacts of two types of regularization.

**Outline.** We provide the related work and background knowledge in Sections 2 and 3, respectively. We begin by interpreting the benefits of distributional learning as uncertainty-aware exploration in *value-based* CDRL in Section 4. We further study this exploration benefit *within the policy gradient framework* in Section 5, where we directly compare it with the vanilla entropy regularization in MaxEnt RL. Extensive experiments demonstrate the regularized exploration benefit of distributional RL and its mutual impact with vanilla entropy regularization in MaxEnt RL in Section 6.

## 2 RELATED WORK

**Distributional Learning via Categorical Representation.** Categorical learning has been widely employed, with advantages in representation (Pan et al., 2019; Jang et al., 2016) and optimization (Imani & White, 2018; Sun et al., 2024a). The empirical superiority of categorical distribution learning has increasingly gained attention in various RL tasks (Farebrother et al., 2024), even beyond the classical category of CDRL. Thus, a pressing need exists to examine the theoretical foundations of categorical distributional learning, particularly in the RL context. The perspective of uncertainty-aware regularization-based exploration that our research introduces adds a significant theoretical understanding of the benefits of being categorical distributional in RL.

**Exploration in RL in the Entropy Principle.** As a general and effective mechanism, the entropy principle has been extensively studied to enhance the exploration in RL. Classical algorithms are established upon the MaxEnt RL framework (Williams & Peng, 1991), including soft Q-learning (Haarnoja et al., 2017), Soft Actor Critic (SAC) (Haarnoja et al., 2018a) and their variants (Han & Sung, 2021). The key characteristic in MaxEnt RL is to directly incorporate the entropy term regarding the policy in the objective function, while other works introduce the variation

in decision-making in distinct ways. These works include (Mavrin et al., 2019), which utilizes the variance of return distribution to promote the exploration, and (Lee et al., 2021), which relies on the ensemble technique. By contrast, we show that distributional learning in RL implicitly encourages a distinct uncertainty-aware exploration driven by optimizing the derived cross-entropy-based regularization that measures the discrepancy between the agent's uncertain estimate and the environment.

## 3 PRELIMINARIES

**Markov Decision Process (MDP) and Classical RL.** An environment is modeled via an Markov Decision Process $(\mathcal{S}, \mathcal{A}, \mathcal{R}, P, \gamma)$, with a set of states $\mathcal{S}$ and actions $\mathcal{A}$, the bounded reward function $\mathcal{R} : \mathcal{S} \times \mathcal{A} \rightarrow \mathcal{P}([R_{\min}, R_{\max}])$, the transition kernel $P : \mathcal{S} \times \mathcal{A} \rightarrow \mathcal{P}(\mathcal{S})$, and a discounted factor $\gamma \in [0, 1]$. We denote the reward the agent receives at time $t$ as $r_t(s_t, a_t) \sim \mathcal{R}(s_t, a_t)$. Given a policy $\pi$, the key quantity of interest is the return $Z^\pi$, which is the total cumulative rewards over the course of a trajectory defined by $Z^\pi(s, a) = \sum_{t=0}^{\infty} \gamma^t r_t | s_0 = s, a_0 = a$. Classical RL focuses on estimating the expectation of the return, i.e., $Q^\pi(s, a) = \mathbb{E}_\pi \left[ \sum_{t=0}^{+\infty} \gamma^t r_t | s_0 = s, a_0 = a \right]$. We also define Bellman evaluation operator $\mathcal{T}^\pi Q(s, a) = \mathbb{E}[\mathcal{R}(s, a)] + \gamma \mathbb{E}_{s' \sim P, a' \sim \pi} [Q(s', a')]$, and Bellman optimality operator $\mathcal{T}^{\mathrm{opt}} Q(s, a) = \mathbb{E}[\mathcal{R}(s, a)] + \gamma \max_{a'} \mathbb{E}_{s' \sim P} [Q(s', a')]$.

**Distributional RL and CDRL.** Instead of only learning the expectation in classical RL, distributional RL models the full distribution of the return random variable $Z^\pi$. The return distribution $\eta^\pi : \mathcal{S} \times \mathcal{A} \rightarrow \mathcal{P}(\mathbb{R})$ is defined as $\eta^\pi(s, a) = \mathcal{D}(Z^\pi(s, a))$, where $\mathcal{D}$ extracts the distribution of the random variable. $\eta^\pi(s, a)$ is updated via the distributional Bellman operator $\mathfrak{T}^\pi$, defined by $\mathfrak{T}^\pi Z(s, a) \overset{D}{=} \mathcal{R}(s, a) + \gamma Z(s', a')$, where $\overset{D}{=}$ implies random variables of both sides are equal in distribution. CDRL (Bellemare et al., 2017a), such as C51, is the first successful distributional RL algorithm family that approximates the return distribution by a discrete categorical distribution $\widehat{\eta}^\pi = \sum_{i=1}^{N} p_i \delta_{z_i}$, where $\{z_i\}_{i=1}^{N}$ is a set of fixed supports and $\{p_i\}_{i=1}^{N}$ are learnable probabilities. The leverage of a heuristic projection operator $\Pi_{\mathcal{C}}$ (see Appendix A for more details) and the Kullback–Leibler (KL) divergence guarantee the theoretical convergence of CDRL under Cramér distance or Wasserstein distance in the tabular setting (Rowland et al., 2018).

## 4 REGULARIZATION BENEFITS IN VALUE-BASED DISTRIBUTION RL

In this section, we simplify value-based distributional RL to a Neural Fitted Z-Iteration (Neural FZI) process in Section 4.1, within which the objective function of distributional learning can be further rewritten as an entropy-regularized form as shown in Section 4.2. Finally, we characterize the role of the derived entropy-based regularization as uncertain-aware regularized exploration in Section 4.3.

### 4.1 DISTRIBUTIONAL RL: NEURAL FZI

**Classical RL: Neural Fitted Q-Iteration (Neural FQI).** Neural FQI (Fan et al., 2020; Riedmiller, 2005) offers a statistical explanation of DQN (Mnih et al., 2015), capturing its key features, including experience replay and the target network $Q_{\theta^*}$. In Neural FQI, we update a parameterized $Q_\theta$ in each iteration $k$ of an iterative regression framework: $Q_\theta^{k+1} = \mathrm{argmin}_{Q_\theta} \frac{1}{n} \sum_{i=1}^{n} \left[ y_i^k - Q_\theta(s_i, a_i) \right]^2$ **(Neural FQI)**, where the target $y_i^k = r(s_i, a_i) + \gamma \max_{a \in \mathcal{A}} Q_{\theta^*}^k(s_i', a)$ is fixed within every $T_{\mathrm{target}}$ steps to update target network $Q_{\theta^*}$ by letting $Q_{\theta^*}^k = Q_\theta^k$. The experience buffer induces independent samples $\{(s_i, a_i, r_i, s_i')\}_{i \in [n]}$. If $\{Q_\theta : \theta \in \Theta\}$ is sufficiently large such that it contains $\mathcal{T}^{\mathrm{opt}} Q_{\theta^*}^k$, i.e., the realizable assumption in learning theory (Mohri, 2018), Neural FQI has the solution $Q_\theta^{k+1} = \mathcal{T}^{\mathrm{opt}} Q_{\theta^*}^k$, which is exactly the updating rule under Bellman optimality operator (Fan et al., 2020).

**Distributional RL: Neural Fitted Z-Iteration (Neural FZI).** While our analysis is not intended to involve properties of neural networks, we interpret distributional RL as Neural FZI as it is by far closest to the practical algorithms. Analogous to Neural FQI, we simplify value-based distributional RL algorithms denoted by the parameterized $Z_\theta$ into Neural FZI, which is formulated as

$$Z_\theta^{k+1} = \underset{Z_\theta}{\mathrm{argmin}} \frac{1}{n} \sum_{i=1}^{n} d_p(Y_i^k, Z_\theta(s_i, a_i)), \tag{1}$$

where we denote the target random variable $Y_i^k = \mathcal{R}(s_i, a_i) + \gamma Z_{\theta^*}^k (s_i', \pi_Z(s_i'))$ with the policy $\pi_Z$ following the greedy rule $\pi_Z(s_i') = \text{argmax}_{a'} \mathbb{E}\left[Z_{\theta^*}^k(s_i', a')\right]$. The target $Y_i^k$ is fixed within every $T_{\text{target}}$ steps to update target network $Z_{\theta^*}$. $d_p$ is a distribution divergence between two distributions, and the lower cases of random variables $s_i'$ and $\pi_Z(s_i')$ are given for convenience in notations.

## 4.2 DISTRIBUTIONAL RL: ENTROPY-REGULARIZED NEURAL FQI

As mentioned previously in preliminary knowledge (Section 3), CDRL employs neural networks to learn the probabilities $\{p_i\}_{i=1}^N$ in a discrete categorical distribution to represent $Z_\theta$, and choose KL divergence as $d_p$ in Eq. 1 of Neural FZI. We next decompose the KL-based distributional loss $d_p$ in CDRL by decomposing an equivalent histogram density estimator $\hat{p}$ in representing $Z_\theta$.

**Return Density Decomposition.** To characterize the impact of additional return distribution knowledge beyond the expectation of $Z^\pi$, we use a variant of *gross error model* from robust statistics (Huber, 2004), which was also similarly utilized to analyze Label Smoothing (Müller et al., 2019) and Knowledge Distillation (Hinton et al., 2015). Akin to the categorical representation in CDRL (Dabney et al., 2018b), we utilize a *histogram function estimator* $\hat{p}^{s,a}(x)$ with $N$ bins to approximate an arbitrary continuous true density $p^{s,a}(x)$ of $Z^\pi(s, a)$, given a state $s$ and action $a$. In contrast to categorical parameterization, which is defined on a set of fixed supports, the histogram estimator operates over a continuous interval, enabling more nuanced analysis within continuous functions. Given a fixed set of supports $l_0 \le l_1 \le ... \le l_N$ with the equal bin size as $\Delta$, each bin is thus dented as $\Delta_i = [l_{i-1}, l_i)$, $i = 1, ..., N-1$ with $\Delta_N = [l_{N-1}, l_N]$. As such, the histogram density estimator is formulated by $\hat{p}^{s,a}(x) = \sum_{i=1}^N p_i \mathbb{1}(x \in \Delta_i)/\Delta$ with $p_i$ as the coefficient in the $i$-th bin $\Delta_i$. Denote $\Delta_E$ as the interval that $\mathbb{E}[Z^\pi(s, a)]$ falls into, i.e., $\mathbb{E}[Z^\pi(s, a)] \in \Delta_E$. See Figure 1 for the illustration of a histogram density function $\hat{p}^{s,a}$. Putting all together, we apply an action-state return density decomposition over the histogram density estimator $\hat{p}^{s,a}$:

$$\hat{p}^{s,a}(x) = (1 - \epsilon)\mathbb{1}(x \in \Delta_E)/\Delta + \epsilon\hat{\mu}^{s,a}(x), \qquad (2)$$

where $\hat{p}^{s,a}$ is decomposed into a single-bin histogram $\mathbb{1}(x \in \Delta_E)/\Delta$ with all mass on $\Delta_E$ and an **induced** histogram density function $\hat{\mu}^{s,a}$ evaluated by $\hat{\mu}^{s,a}(x) = \sum_{i=1}^N p_i^\mu \mathbb{1}(x \in \Delta_i)/\Delta$ with $p_i^\mu$ as the coefficient of the $i$-th bin $\Delta_i$. $\epsilon$ is a hyper-parameter before the decomposition, controlling the proportion between $\mathbb{1}(x \in \Delta_E)/\Delta$ and $\hat{\mu}^{s,a}(x)$. More specifically, the induced histogram density function $\hat{\mu}^{s,a}$ in the second term of Eq. 2 represents the difference between the full histogram function $\hat{p}^{s,a}$ and a single-bin histogram, which only captures the mean. This difference indicates that $\hat{\mu}^{s,a}$ captures the distribution information beyond its expectation

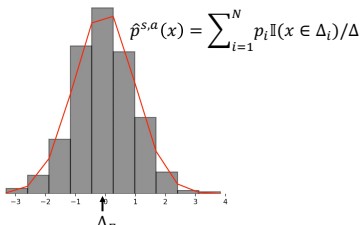

Figure 1: Histogram Estimator.

$\mathbb{E}[Z^\pi(s, a)]$, incorporating higher-moments information. The reflects the influence of using full distribution on the performance of distributional RL. The additional leverage of $\hat{\mu}^{s,a}$ in the distributional loss explains the behavior differences between classical and distribution RL algorithms. We first demonstrate that $\hat{\mu}^{s,a}$ is a valid probability density function under certain $\epsilon$ in Proposition 1.

**Proposition 1.** *(Decomposition Validity) Denote $\hat{p}^{s,a}(x \in \Delta_E) = p_E/\Delta$, where $p_E$ is the coefficient on the bin $\Delta_E$. $\hat{\mu}^{s,a}(x) = \sum_{i=1}^N p_i^\mu \mathbb{1}(x \in \Delta_i)/\Delta$ is a valid density if and only if $\epsilon \ge 1 - p_E$.*

The proof can be found in Appendix B. Proposition 1 demonstrates that the return density decomposition is valid when the hyper-parameter $\epsilon$ is well specified as $\epsilon \ge 1 - p_E$. Under this condition, our analysis maintains the standard categorical distributional framework in distributional RL.

**Remark: Equivalence between Histogram Density Estimator and Categorical Representation.** The histogram function is a continuous estimator in contrast to the discrete nature of categorical parameterization. We show that they are equivalent in representing a density in Appendix C. As a supplementary analysis, with attribution to (Wasserman, 2006), we also discuss necessary theoretical underpinnings of the histogram density estimator in the context of distributional RL in Appendix D.

**Distributional RL: Entropy-regularized Neural FQI.** We apply the decomposition in Eq. 2 on the histogram density function, denoted as $\hat{p}^{s_i', \pi_Z(s_i')}$, of the target return $Y_i^k = \mathcal{R}(s_i, a_i) + \gamma Z_{\theta^*}^k(s_i', \pi_Z(s_i'))$ in Eq. 1 of Neural FZI. Consequently, we have $\hat{p}^{s_i', \pi_Z(s_i')}(x) = (1 - \epsilon)\mathbb{1}(x \in$

$\Delta_E^i)/\Delta + \epsilon\widehat{\mu}^{s_i',\pi_Z(s_i')}(x)$, where $\Delta_E^i$ represents the interval that the expectation of the target return $Y_i^k$ falls into, i.e., $\mathbb{E}\left[Y_i^k\right] \in \Delta_E^i$, and $\widehat{\mu}_i^{s_i',\pi_Z(s_i')}$ is the induced histogram density function, similar to the role of $\widehat{\mu}^{s,a}$ in Eq. 2. Let $\mathcal{H}(U,V)$ be the cross-entropy between two probability measures $U$ and $V$, i.e., $\mathcal{H}(U,V) = -\int_{x\in\mathcal{X}} U(x)\log V(x)\,\mathrm{d}x$. Immediately, we can derive the following entropy-regularized loss function form of Neural FZI for distributional RL in Proposition 2, with the proof provided in Appendix F.

**Proposition 2.** *(Decomposed Neural FZI) Denote* $q_\theta^{s,a}$ *as the histogram estimator of* $Z_\theta^k(s,a)$ *in Neural FZI. Based on Eq. 2 and the KL divergence as* $d_p$*, Neural FZI in Eq. 1 is simplified as*

$$Z_\theta^{k+1} = \underset{q_\theta}{\arg\min}\frac{1}{n}\sum_{i=1}^n[\underbrace{-\log q_\theta^{s_i,a_i}(\Delta_E^i)}_{(a)} + \alpha\mathcal{H}(\widehat{\mu}^{s_i',\pi_Z(s_i')}, q_\theta^{s_i,a_i})], \tag{3}$$

*where* $\alpha = \varepsilon/(1-\varepsilon) > 0$ *and the term (a) is negative log-likelihood function centered on* $\Delta_E^i$*.*

**Connection between Neural FQI and FZI.** A crucial bridge between classical and distributional RL is established in Proposition 3, where we show that minimizing the term (a) in Eq. 3 of Neural FZI is asymptotically equivalent to minimizing Neural FQI in terms of the minimizers. As such, the regularization term $\alpha\mathcal{H}(\widehat{\mu}^{s_i',\pi_Z(s_i')}, q_\theta^{s_i,a_i})$ interprets the potential benefits of CDRL over classical RL. For the uniformity of notation, we still use $s, a$ in the following analysis instead of $s_i, a_i$.

**Proposition 3.** *(Equivalence between **the term (a)** in Decomposed Neural FZI and Neural FQI) In Eq. 3 of Neural FZI, assume the function class* $\{Z_\theta : \theta \in \Theta\}$ *is sufficiently large such that it contains the target* $\{Y_i^k\}_{i=1}^n$ *for all* $k$*, when* $\Delta \to 0$*, minimizing **the term (a)** in Eq. 3 implies*

$$\mathbb{P}(Z_\theta^{k+1}(s,a) = \mathcal{T}^{opt}Q_{\theta*}^k(s,a)) = 1, \tag{4}$$

*where* $\mathcal{T}^{opt}Q_{\theta*}^k(s,a)$ *is the scalar-valued target in the k-th phase of Neural FQI.*

See Appendix G for the detailed proof. Proposition 3 demonstrates that as $\Delta \to 0$, the random variable $Z_\theta^{k+1}(s,a)$ with the limiting distribution in Neural FZI (distributional RL) will *degrade* to a constant $\mathcal{T}^{\text{opt}}Q_{\theta*}^k(s,a)$, the minimizer (scalar-valued target) in Neural FQI (classical RL). That being said, *minimizing the term (a) in Neural FZI is asymptotically equivalent to minimizing Neural FQI with the same limiting minimizer*. A formal proof for convergence in distribution with the rate $o(\Delta)$ is given in Appendix G. With the connection between optimizing the term (a) of Neural FZI with Neural FQI in Proposition 3, we can leverage the regularization term $\alpha\mathcal{H}(\widehat{\mu}^{s_i',\pi_Z(s_i')}, q_\theta^{s_i,a_i})$ to explain the potential superiority of CDRL over classical RL. The realizable assumption that $\{Z_\theta : \theta \in \Theta\}$ is sufficiently large such that it contains $\{Y_i^k\}_{i=1}^n$ implies good in-distribution generalization performance in each phase of Neural FZI, which is commonly used in analyzing distributional RL, e.g., (Wu et al., 2023). This connection is also consistent with the mean-preserving property of distributional RL in the tabular setting (Rowland et al., 2018), but we extend this conclusion to the arbitrary function approximation setting using a histogram density estimator.

### 4.3 UNCERTAINTY-AWARE REGULARIZED EXPLORATION

Thanks to the equivalence between the term (a) of decomposed Neural FZI and FQI, the behavior difference of distributional RL as opposed to classical RL is thus attributed to the second regularization term $\alpha\mathcal{H}(\widehat{\mu}^{s_i',\pi_Z(s_i')}, q_\theta^{s_i,a_i})$. Minimizing Neural FZI pushes $q_\theta^{s,a}$ for the current return density estimator to catch up with the target return density function of $\widehat{\mu}^{s_i',\pi_Z(s_i')}$, which encompasses the uncertainty of the whole return distribution in the learning course beyond only its expectation. Since it is a prevalent notion that distributional RL can significantly reduce intrinsic uncertainty of the environment (Mavrin et al., 2019; Dabney et al., 2018a), the derived distribution-matching regularization term $\alpha\mathcal{H}(\widehat{\mu}^{s_i',\pi_Z(s_i')}, q_\theta^{s_i,a_i})$ helps to capture more uncertainty of the environment by modeling the whole return distribution beyond the expectation. In Section 5, we show that this derived regularization contributes to *uncertainty-aware regularized exploration* in the policy optimization.

**Remark: Approximation of** $\widehat{\mu}^{s',\pi_Z(s')}$**.** In practical distributional RL algorithms, we typically use temporal-difference (TD) learning to attain the target probability density estimate $\widehat{\mu}^{s',\pi_Z(s')}$ based on Eq. 2, provided $\mathbb{E}\left[Z(s,a)\right]$ exists and $\epsilon \geq 1 - p_E$ in Proposition 1. The approximation error

of $\widehat{\mu}^{s',\pi_Z(s')}$ is fundamentally determined by the TD learning nature. A desirable approximation of $\widehat{\mu}^{s',\pi_Z(s')}$ intuitively leads to performance improvement in distributional RL. As KL divergence is used in CDRL, we also discuss the usage of KL divergence in distributional RL in Appendix E.

## 5 REGULARIZATION BENEFITS IN ACTOR CRITIC FRAMEWORK

### 5.1 CONNECTION WITH MAXENT RL

**Motivation for the Connection.** The maximum entropy regularization is commonly used in RL, which has various conceptual and practical advantages. Firstly, the learned policy is encouraged to visit states with high entropy in the future, promoting the exploration of diverse actions (Han & Sung, 2021; Haarnoja et al., 2018a; Williams & Peng, 1991). It also considerably improves the learning speed (Mei et al., 2020) and therefore is widely employed in state-of-the-art algorithms, e.g., Soft Actor-Critic (SAC) (Haarnoja et al., 2018a). Similar empirical benefits of both distributional RL and MaxEnt RL motivate us to probe their underlying connection, especially in exploration.

**Explicit Entropy Regularization in MaxEnt RL.** MaxEnt RL (Williams & Peng, 1991) *explicitly* encourages exploration by optimizing for policies to reach states with higher entropy in the future:

$$J(\pi) = \sum_{t=0}^{T} \mathbb{E}_{(s_t,a_t)\sim\rho_\pi} \left[ r\left(s_t,a_t\right) + \beta\mathcal{H}(\pi(\cdot|s_t)) \right], \tag{5}$$

where $\mathcal{H}\left(\pi_\theta\left(\cdot|s_t\right)\right) = -\sum_a \pi_\theta\left(a|s_t\right)\log\pi_\theta\left(a|s_t\right)$ and $\rho_\pi$ is the generated distribution following $\pi$. The temperature parameter $\beta$ determines the relative importance of the entropy term against the cumulative rewards and thus controls the action diversity of the optimal policy learned via Eq. 5.

**Implicit Entropy Regularization in Distributional RL.** For a direct comparison with MaxEnt RL, it is required to specifically analyze the impact of the regularization term in Eq. 3. Consequently, we incorporate the distribution-matching regularization of distributional RL into the Actor Critic (AC) framework akin to MaxEnt RL, enabling us to consider a new soft Q-value. The new Q function can be computed iteratively by applying a modified Bellman operator denoted as $\mathcal{T}_d^\pi$, called *Distribution-Entropy-Regularized Bellman Operator*. Given a fixed $q_\theta$, $\mathcal{T}_d^\pi$ is defined as

$$\mathcal{T}_d^\pi Q\left(s_t,a_t\right) \triangleq r\left(s_t,a_t\right) + \gamma\mathbb{E}_{s_{t+1}\sim P(\cdot|s_t,a_t)}\left[V\left(s_{t+1}|s_t,a_t\right)\right], \tag{6}$$

where a new soft value function $V\left(s_{t+1}|s_t,a_t\right)$ conditioned on $s_t,a_t$ is defined by

$$V\left(s_{t+1}|s_t,a_t\right) = \mathbb{E}_{a_{t+1}\sim\pi}\left[Q\left(s_{t+1},a_{t+1}\right)\right] + f(\mathcal{H}\left(\mu^{s_t,a_t},q_\theta^{s_t,a_t}\right)), \tag{7}$$

where $f$ is a continuous increasing function over the cross-entropy $\mathcal{H}$. $\mu^{s_t,a_t}$ is the induced true target return histogram density function via the decomposition in Eq. 2 regardless of its expectation, which can be approximated via bootstrap TD estimate $\widehat{\mu}^{s_{t+1},\pi_Z(s_{t+1})}$ similar to Eq. 3. In this specific tabular setting regarding $s_t,a_t$, we particularly use $q_\theta^{s_t,a_t}$ to approximate the true density function of $Z(s_t,a_t)$. The $f$ transformation over the cross-entropy $\mathcal{H}$ between $\mu^{s_t,a_t}$ and $q_\theta^{s_t,a_t}(x)$ serves as the uncertainty-aware entropy regularization that we implicitly derive from value-based distributional RL in Section 4.2. By optimizing $q_\theta$, the value-based critic component

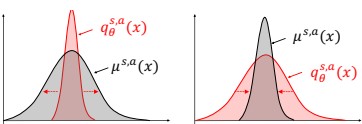

Figure 2: $q_\theta^{s,a}$ is optimized to disperse (left) or concentrate (right) to align with the uncertainty of target return distributions.

in Actor-Critic, this regularization reduces the mismatch between the target return distribution and current estimate, aligning with the regularization effect analyzed in Section 4.3. As illustrated in Figure 2, $q_\theta^{s,a}$ is optimized to **catch up with** the uncertainty of the target return distribution of $\mu^{s,a}$, expanding the knowledge of algorithms about the environment uncertainty for more informative decisions. Next, we elaborate on its additional impact on policy learning in the actor-critic in contrast to MaxEnt RL.

**Reward Augmentation for Policy Learning.** As opposed to the vanilla entropy regularization in MaxEnt RL that explicitly encourages the policy to explore, our derived distribution-matching regularization in distributional RL plays a role of **reward augmentation** for policy learning. Compared with classical RL, the augmented reward incorporates additional return distribution knowledge in

the learning process. As we will show later, *the augmented reward encourages policies to reach states $s_t$ with actions $a_t \sim \pi(\cdot|s_t)$, whose current action-state return distribution $q_\theta^{s_t,a_t}$* **lags far behind** *the target one, measured by the magnitude of cross entropy.*

For a detailed comparison with MaxEnt RL, we now focus on the properties of our distribution-matching regularization in the AC framework. In Lemma 1, we first show that our Distribution-Entropy-Regularized Bellman operator $\mathcal{T}_d^\pi$ still inherits the convergence property in the policy evaluation phase with a cumulative augmented reward function as the new objective function $J'(\pi)$.

**Lemma 1.** *(Distribution-Entropy-Regularized Policy Evaluation) Consider the distribution-entropy-regularized Bellman operator $\mathcal{T}_d^\pi$ in Eq. 6 and assume $\mathcal{H}(\mu^{s_t,a_t}, q_\theta^{s_t,a_t})$ is bounded for all $(s_t, a_t) \in \mathcal{S} \times \mathcal{A}$. We define $Q^{k+1} = \mathcal{T}_d^\pi Q^k$. Given $q_\theta$, $Q^{k+1}$ will converge to a corrected Q-value of $\pi$ as $k \to \infty$ with the new objective function $J'(\pi)$ defined as*

$$J'(\pi) = \sum_{t=0}^{T} \mathbb{E}_{(s_t,a_t)\sim\rho_\pi} \left[ r\left(s_t, a_t\right) + \gamma f(\mathcal{H}\left(\mu^{s_t,a_t}, q_\theta^{s_t,a_t}\right)) \right]. \tag{8}$$

We remain the updating rule $\pi_{\text{new}} = \arg\max_{\pi'\in\Pi} \mathbb{E}_{a_t\sim\pi'}\left[Q^{\pi_{\text{old}}}(s_t, a_t)\right]$ in policy improvement. Next, we derive a new policy iteration algorithm, called *Distribution-Entropy-Regularized Policy Iteration (DERPI)*, alternating between policy evaluation in Eq. 6 and policy improvement. It provably converges to a policy regularized by the distribution-matching term in Theorem 1.

**Theorem 1.** *(Distribution-Entropy-Regularized Policy Iteration) Repeatedly applying distribution-entropy-regularized policy evaluation in Eq. 6 and the policy improvement, the policy converges to an optimal policy $\pi^*$ such that $Q^{\pi^*}\left(s_t, a_t\right) \geq Q^\pi\left(s_t, a_t\right)$ for all $\pi \in \Pi$.*

Please refer to Appendix H for the proof of Lemma 1 and Theorem 1. Theorem 1 demonstrates that if we incorporate the distribution-matching regularization into the policy gradient framework in Eq. 8, we can design a variant of "soft policy iteration" (Haarnoja et al., 2018a) that can guarantee the convergence to an optimal policy given any fixed $q_\theta$. While our theoretical analysis adheres to the standard analytical framework in MaxEnt RL, we finally recognize a fundamental difference between our decomposed entropy regularization and the vanilla entropy regularization in MaxEnt RL. Next, we summarize the distinct regularized exploration effects of MaxEnt RL and CDRL.

**Uncertainty-aware Regularized Exploration in CDRL Compared with MaxEnt RL.** For the objective function $J(\pi)$ in Eq. 5 of MaxEnt RL, the state-wise entropy $\mathcal{H}(\pi(\cdot|s_t))$ is maximized explicitly *w.r.t.* $\pi$ for policies with a higher entropy in terms of diverse actions to encourage an explicit exploration. For the objective function $J'(\pi)$ in Eq. 8 of distributional RL, the policy $\pi$ is implicitly optimized through **the action selection $a_t \sim \pi(\cdot|s_t)$ mechanism** guided by an augmented reward signal from the distribution-matching regularization $f(\mathcal{H}\left(\mu^{s_t,a_t}, q_\theta^{s_t,a_t}\right))$. Concretely, the learned policy is encouraged to visit state $s_t$ along with the policy-determined action via $a_t \sim \pi(\cdot|s_t)$, whose current action-state return distributions $q_\theta^{s_t,a_t}$ *lag far behind* the target return distributions. This discrepancy is measured by the magnitude of the cross entropy between two return distributions. A large discrepancy indicates that the uncertainty of current return distribution is considerably misestimated for considered states, promoting an uncertainty-aware exploration against these states in policy optimization. This also indicates that the policy learning in CDRL is additionally driven by the uncertainty difference between the current and the target estimates, leading to a distinct exploration strategy of distributional RL compared with MaxEnt RL.

**Interplay of Uncertainty-aware Regularization in Distributional Actor-Critic.** Putting the critic and actor learning together in distributional RL, we reveal their interplay impact of the uncertainty-aware regularized exploration when compared with expectation-based RL: 1) on the one hand, the actor (policy) learning seeks states and actions whose current return distribution estimate lags far behind the true one determined by the environment, 2) on the other hand, the critic learning reduces the return distribution mismatch on the states and actions explored by the actor or the policy, as illustrated in Figure 2. This uncertainty-aware exploration effect arises from the derived regularization after the return density decomposition, interpreting the benefits of CDRL over classical RL.

## 5.2 DERAC ALGORITHM: INTERPOLATING AC AND DISTRIBUTIONAL AC

**Motivation.** The convergence guarantee of DERPI given a fixed $q_\theta$ in Section 5.1 provides sufficient insights to understand the uncertainty-aware regularized exploration. To further substantiate

the validity of introducing the decomposed entropy into the actor-critic **with the general function approximation**, we extend DERPI into a practical algorithm with favorable interpretability. Unlike SAC, which introduces another value function network, we only parameterize the return distribution $q_\theta(s_t, a_t)$ and the policy $\pi_\phi(a_t|s_t)$, where we use $\mathbb{E}[q_\theta]$ to represent the Q function without parameterizing it again. Remarkably, the resulting *Distribution-Entropy-Regularized Actor-Critic (DERAC)* algorithm can interpolate expectation-based AC and distributional AC.

**Optimize the critic** $q_\theta$. The new value function $\hat{J}_q(\theta)$ is originally trained to minimize the squared residual error of Eq. 6. We show that $\hat{J}_q(\theta)$ can be simplified as:

$$\hat{J}_q(\theta) \propto (1-\lambda)\mathbb{E}_{s,a}\left[(\mathcal{T}^\pi\mathbb{E}[q_{\theta^*}(s,a)] - \mathbb{E}[q_\theta(s,a)])^2\right] + \lambda\mathbb{E}_{s,a}[\mathcal{H}(\mu^{s,a}, q_\theta^{s,a})], \qquad (9)$$

where we use a particular increasing function $f(\mathcal{H}) = (\tau\mathcal{H})^{\frac{1}{2}}/\gamma$ and $\lambda = \frac{\tau}{1+\tau} \in [0, 1], \tau \geq 0$ is the hyperparameter that controls the uncertainty-aware regularization effect. The proof is given in Appendix I. Interestingly, when we leverage the whole target density function $\widehat{p}^{s,a}$ to approximate the true return distribution of $\mu^{s,a}$, the objective function in Eq. 9 can be viewed as an exact interpolation of loss functions between expectation-based AC (the first term) and categorical distributional AC loss (the second term) (Ma et al., 2020). In our implementation, for the target $\mathcal{T}^\pi\mathbb{E}[q_{\theta^*}(s,a)]$, we use the target return distribution neural network $q_{\theta^*}$ to stabilize the training, which is consistent with the Neural FZI framework analyzed in Section 4.1.

**Optimize the policy** $\pi_\phi$. We optimize $\pi_\phi$ in the policy optimization based on the Q-function and therefore the new objective function $\hat{J}_\pi(\phi)$ can be expressed as $\hat{J}_\pi(\phi) = \mathbb{E}_{s,a\sim\pi_\phi}[\mathbb{E}[q_\theta(s,a)]]$. The complete DERAC algorithm is presented in Algorithm 2 of Appendix K.

**Remark on DERAC and Its Difference from Categorical Distributional AC.** The careful neural architecture design and selection of the function $f$ endow the loss function of DERAC with interpretability. However, the DERAC algorithm is not our main focus but primarily serves to substantiate the efficacy of the uncertainty-aware regularized exploration in distributional RL within an actor-critic framework, rather than to achieve superior real-world performance. In contrast to Categorical Distributional AC, which depends entirely on distributional learning in policy optimization, DERAC interpolates between expectation-based and distributional learning. In Section 6.2, we empirically demonstrate that this interpolation form can be more suitable in specific environments than distributional AC, helping to *mitigate the excessive exploration* in fully distributional learning.

# 6 EXPERIMENTS

We provide a comprehensive demonstration of our theoretical analysis using both Atari games and MuJoCo environments. In Section 6.1, we first verify that the uncertainty-aware regularization controls the performance benefit of CDRL by varying $\epsilon$ in the return density decomposition. In Section 6.2, we examine the interpolation performance of the proposed DERAC algorithm in continuous control environments to substantiate the uncertain-aware regularized exploration in actor-critic algorithms. Finally, we explore the mutual impacts between the vanilla entropy regularization in MaxEnt RL and the uncertainty-aware one from CDRL in Section 6.3, with a slight extension to quantile-based distributional RL, e.g., Implicit Quantile Networks (IQN) (Dabney et al., 2018a). More implementation details, including the description of baselines, are provided in Appendix J.

## 6.1 REGULARIZATION EFFECT BY VARYING $\epsilon$ IN RETURN DENSITY DECOMPOSITION

We demonstrate the decomposed uncertainty-aware entropy regularization, which is derived in Eq. 3 through the return density function decomposition, plays a crucial role in the empirical outperformance of CDRL over classical RL. Our experiments are conducted on both typical Atari games and Mujoco environments. Particularly, for the categorical distributional loss in C51 or the critic loss in the actor-critic algorithms, we replace the whole target histogram density $\widehat{p}^{s,a}$ with the derived $\widehat{\mu}^{s,a}$ decomposed under different $\varepsilon$ based on Eq. 2. **We then employ $\widehat{\mu}^{s,a}$ instead of $\widehat{p}^{s,a}$ as the target return distribution in the distributional loss of CDRL, leading to the decomposed algorithms, denoted by $\mathcal{H}(\mu, q_\theta)$.** This decomposed algorithm enables us to assess the uncertainty-aware regularization effect of distributional RL by comparing its performance with the classical RL and CDRL. To ensure a pre-specified $\epsilon$ that guarantees a valid decomposition analyzed in Proposition 1, we use a

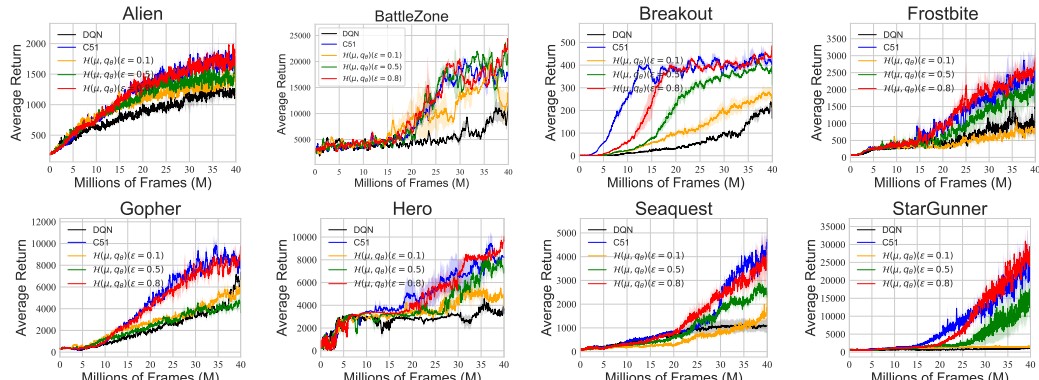

Figure 3: Learning curves of value-based CDRL, i.e., C51 algorithm, and the decomposed algorithm $\mathcal{H}(\mu, q_\theta)$ after the return distribution decomposition with different $\varepsilon$ on eight Atari games. Results are averaged over 3 seeds and the shade represents the standard deviation.

new notation $\varepsilon$, which shares the same utility with $\epsilon$ and is more convenient in the implementation. $\varepsilon$ is defined as the mass proportion centered at the bin that contains the expectation *when transporting the mass to other bins*. A large proportion probability $\varepsilon$, which transports less mass to other bins, corresponds to a large $\epsilon$ in Eq. 2. Increasing $\varepsilon$ indicates that the decomposed algorithm performs more similarly to a pure CDRL algorithm. See Appendix J.2 for more explanation, including the transformation equation between $\epsilon$ and $\varepsilon$, and the decomposition details of our $\mathcal{H}(\mu, q_\theta)$ algorithm.

Figure 3 showcases that as $\varepsilon$ gradually decreases from 0.8 to 0.1, learning curves of decomposed C51, denoted as $\mathcal{H}(\mu, q_\theta)(\varepsilon = 0.8/0.5/0.1)$, tend to degrade from vanilla C51 to DQN across most Atari games. The sensitivity of decomposed algorithm $\mathcal{H}(\mu, q_\theta)$ in terms of $\varepsilon$ depends on the environment. Similar results in continuous control environments can be found in Appendix L.1. Overall, our empirical result corroborates that the decomposed uncertainty-aware entropy regularization from the categorical distributional loss is pivotal to the empirical benefits of CDRL over classical RL.

## 6.2 INTERPOLATION BEHAVIOR OF DERAC: MITIGATING THE EXCESSIVE EXPLORATION

Figure 4 suggests that DERAC (green) converges and tends to "interpolate" between the expectation-based AC and distributional AC denoted by DAC (C51), substantiating the theoretical convergence of the tabular DERPI algorithm in Theorem 1. We highlight that *the primary purpose of introducing DERAC is to interpret the benefits of CDRL from the perspective of uncertain-aware regularized exploration, rather than to pursue the empirical superiority*. In **Group 1**, it is essential to note that DERAC achieves superior performance over both AC and DAC (C51) on bipedalwalkerhardcore, verifying that the interpolation has extra advantages. We posit that the interpolation nature of DE-RAC mitigates the over-exploration when adopting the purely categorical distributional learning in C51, as a pure CDRL algorithm may put too much emphasis on the uncertainty-aware exploration, i.e., all weight on the regularization term in Entropy-regularized Neural FQI in Eq. 3. In **Group 2**

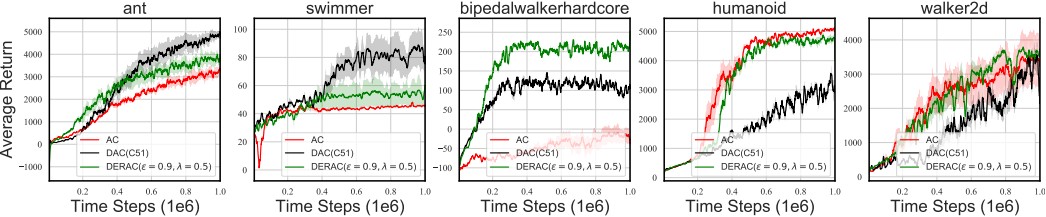

Figure 4: Learning curves of DERAC over 5 seeds on MuJoCo. No vanilla entropy regularization is used in AC or DAC. **Group 1**: Ant, Swimmer and Bipedalwalkerhardcore, where DAC (C51) outperforms AC. **Group 2**: Humanoid and Walker2d, where AC outperforms DAC (C51).

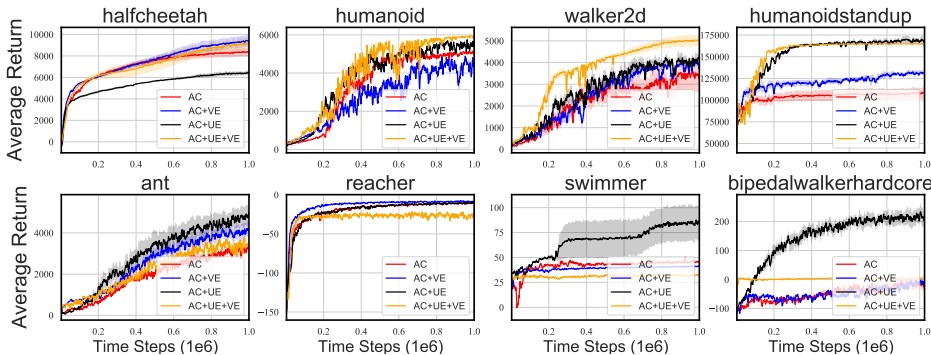

Figure 5: Learning curves of *AC*, *AC+VE* (SAC), **AC+UE** (DAC) and *AC+UE+VE* (DSAC) over five seeds across eight MuJoCo environments where DAC and DSAC are based on IQN. (**First Row**): Mutual improvement. (**Second Row**): Potential interference.

where DAC is inferior to AC, it exhibits that DERAC performs similarly to or slightly excels at AC. These results demonstrate that DERAC is more robust and can even surpass DAC (C51) by potentially mitigating the over-exploration of pure distributional RL. Unlike fully distributional RL, which put more weights on uncertainty-aware regularized exploration, DERAC offers a more optimal balance between exploration and exploitation, potentially resulting in better performance in certain environments. We also provide a sensitivity analysis of DERAC regarding $\lambda$ in Appendix L.2.

### 6.3 MUTUAL IMPACTS OF VANILLA ENTROPY REGULARIZATION IN MAXENT RL AND UNCERTAINTY-AWARE REGULARIZATION IN DISTRIBUTIONAL RL

We demonstrate that the two types of regularized exploration encouraged either by **Vanilla Entropy** (**VE**) in MaxEnt RL or **Uncertainty-aware Entropy** (**UE**) in CDRL play distinct roles in the policy learning when used simultaneously, including mutual improvement or potential interference. DSAC stands for Distributional SAC, as initially introduced by (Ma et al., 2020). We perform an ablation study for both DSAC (C51) and DSAC (IQN), where the latter is used to heuristically examine the mutual impacts in quantile-based distributional RL. We present results on DSAC (IQN) and leave similar results on DSAC (C51) in Appendix L.3. Specifically, we denote SAC with/without vanilla entropy as *AC+VE* and *AC*, and Distributional SAC with/without vanilla entropy as *AC+UE+VE* and **AC+UE** or *DAC*. The implementation details can be found in Appendix J.

In the first row in Figure 5, simultaneously employing uncertainty-aware and vanilla entropy regularization renders a mutual improvement. Conversely, the two kinds of regularizations when adopted together lead to performance degradation in the second row in Figure 5, such as Swimmer and Reacher, where *AC+UE+VE* is significantly inferior to **AC+UE** or *AC+VE*. We posit that the potential interference may result from distinct exploration directions in the policy learning for the two types of regularizations. SAC optimizes the policy to visit states with high entropy, while distributional RL updates the policy to explore states and the associated actions whose current return distribution estimate lags far behind the correct one determined by the environment uncertainty.

## 7 DISCUSSIONS AND CONCLUSION

In this paper, we interpret the benefits of CDRL over classical RL as uncertainty-aware regularization derived through the return density decomposition. In contrast to encouraging diverse actions for the exploration in MaxEnt RL, the uncertainty-aware regularization in CDRL promotes to explore states where the environment uncertainty is largely underestimated. This novel perspective from the exploration explains the benefits of (categorical) distributional learning in RL.

**Limitations and Future Work.** The uncertainty-aware regularization with the exploration effect is founded on CDRL. However, it remains elusive whether extending the uncertainty-aware exploration in CDRL to general distributional RL is feasible, given that the analytical techniques in other classes, such as QR-DQN, are highly different from CDRL. We leave this extension as future work.

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

# Appendix

## Table of Contents

# A    CONVERGENCE GUARANTEE OF CATEGORICAL DISTRIBUTIONAL RL AND A DETAILED DESCRIPTION OF C51

**Convergence Properties of CDRL.** Categorical Distributional RL (Bellemare et al., 2017a) uses the heuristic projection operator $\Pi_\mathcal{C}$, which was defined as

$$\Pi_\mathcal{C}(\delta_y) = \begin{cases} \delta_{z_1} & y \leq z_1 \\ \frac{z_{i+1}-y}{z_{i+1}-z_i}\delta_{l_i} + \frac{y-z_i}{z_{i+1}-z_i}\delta_{z_{i+1}} & z_i < y \leq z_{i+1} \\ \delta_{z_N} & y > z_N \end{cases}, \tag{10}$$

After applying the distributional Bellman operator $\mathfrak{T}^\pi$ on the current return distribution $\eta^\pi(s,a)$ in each update, the resulting new distribution, , which we denote as $\widetilde{\eta}^\pi(s,a)$, typically no longer lies in the same (discrete) support with the original one on $\{z_i\}_{i=1}^N$. To maintain the same support, the underpinning of the KL divergence, CDRL additionally applies the projection operator $\Pi_\mathcal{C}$ on the new distribution $\widetilde{\eta}^\pi(s,a)$. This projection rule distributes the weight of $\delta_y$ across the original support points $\{z_i\}_{i=1}^N$ based on the linear interpolation. For example, if $y$ lies in between two support points $z_i$ and $z_{i+1}$, the probability mass on $y$ is split between $z_i$ and $z_{i+1}$ with the weight inversely proportional to its distance ratio to $z_i$ and $z_{i+1}$. Therefore, the projection extends affinely to finite mixtures of Dirac measures, such that for a mixture of Diracs $\sum_{i=1}^N p_i\delta_{y_i}$, we have $\Pi_\mathcal{C}\left(\sum_{i=1}^N p_i\delta_{y_i}\right) = \sum_{i=1}^N p_i\Pi_\mathcal{C}(\delta_{y_i})$. The Cramér distance was recently studied as an alternative to the Wasserstein distances in the context of generative models (Bellemare et al., 2017b). Recall the definition of Cramér distance in the following.

**Definition 1.** *(Definition 3 (Rowland et al., 2018)) The Cramér distance $\ell_2$ between two distributions $\nu_1, \nu_2 \in \mathscr{P}(\mathbb{R})$, with cumulative distribution functions $F_{\nu_1}, F_{\nu_2}$ respectively, is defined by:*

$$\ell_2(\nu_1, \nu_2) = \left(\int_\mathbb{R} (F_{\nu_1}(x) - F_{\nu_2}(x))^2 \ dx\right)^{1/2}.$$

*Further, the supremum-Cramér metric $\bar{\ell}_2$ is defined between two distribution functions $\eta, \mu \in \mathscr{P}(\mathbb{R})^{\mathcal{X} \times \mathcal{A}}$ by*

$$\bar{\ell}_2(\eta, \mu) = \sup_{(x,a) \in \mathcal{X} \times \mathcal{A}} \ell_2\left(\eta^{(x,a)}, \mu^{(x,a)}\right).$$

Thus, the contraction of categorical distributional RL can be guaranteed under Cramér distance:

**Proposition 4.** *(Proposition 2 (Rowland et al., 2018)) The operator $\Pi_\mathcal{C}\mathcal{T}^\pi$ is a $\sqrt{\gamma}$-contraction in $\bar{\ell}_2$.*

An insight behind this conclusion is that Cramér distance endows a particular subset with a notion of orthogonal projection, and the orthogonal projection onto the subset is exactly the heuristic projection $\Pi_\mathcal{C}$ (Proposition 1 in (Rowland et al., 2018)). Rowland et al. (2018) also states that the operator $\Pi_\mathcal{C}\mathcal{T}^\pi$ is contractive under Wasserstein distance.

**Description of CDRL Algorithm, e.g., C51.** With $N = 51$, C51 instantiates the CDRL algorithm. To elaborate the algorithm, we first introduce the pushforward measure $f_\#\nu \in \mathcal{P}(\mathbb{R})$ from Definition

---

**Algorithm 1** CDRL Update (Adapted from Algorithm 1 in (Rowland et al., 2018))

**Require**: Number of atoms $N$, e.g., $N = 51$ in C51, the categorical distribution $\widehat{\eta}(s,a) = \sum_{i=1}^N p_i^{s,a}\delta_{z_i}$ for the current return distribution.
**Input**: Sample transition $(s, a, r, s')$
1: **if** Policy evaluation: **then**
2:     $a^* \sim \pi(\cdot|s')$
3: **else if** Control: **then**
4:     $a^* \leftarrow \arg\max_{a' \in \mathcal{A}} \mathbb{E}_{R \sim \widehat{\eta}(s',a')}[R]$
5: **end if**
6: $\widetilde{\eta}(s,a) \leftarrow (f_{r,\gamma})_\#\widehat{\eta}(s',a^*)$ # Distributional Bellmen update by applying $\widehat{\mathfrak{T}}^\pi$
7: $\widehat{\eta}_{\text{target}}(s,a) \leftarrow \Pi_\mathcal{C}\widetilde{\eta}(s,a)$ # Project target support points onto the original support
**Output**: Compute the distributional loss $\text{KL}(\widehat{\eta}_{\text{target}}(s,a)||\widehat{\eta}(s,a))$ # Choose KL divergence as $d_p$

---

1 in (Rowland et al., 2018). This pushforward measure shifts the support of the probability measure $\mu$ according to the map $f$, which is commonly used in distributional RL literature. In particular, we consider an affine shift map $f_{r,\gamma} : \mathbb{R} \to \mathbb{R}$, defined by $f_{r,\gamma}(x) = r + \gamma x$. As Algorithm 1 displays, we first apply the pushforward measure on the target return distribution $\widehat{\eta}(s', a^*)$ by affinely shifting its support points, leading to a new distribution $\widetilde{\eta}(s, a)$. Next, we project the support points of $\widetilde{\eta}(s, a)$ by employing $\Pi_{\mathcal{C}}$ onto the original support, allowing to compute the KL divergence in the end. Notably, we decompose the distributional objective function on the KL loss $\mathrm{KL}(\widehat{\eta}_{\mathrm{target}}(s, a) \| \widehat{\eta}(s, a))$.

## B  PROOF OF PROPOSITION 1

**Proposition 1**.(Decomposition Validity) Denote $\widehat{p}^{s,a}(x \in \Delta_E) = p_E/\Delta$, where $p_E$ is the coefficient on the bin $\Delta_E$. $\widehat{\mu}^{s,a}(x) = \sum_{i=1}^{N} p_i^{\mu} \mathbb{1}(x \in \Delta_i)/\Delta$ is a valid density if and only if $\epsilon \geq 1 - p_E$.

*Proof.* Recap a valid probability density function requires non-negative and one-bounded probability in each bin and all probabilities should sum to 1.

**Necessity.** (1) When $x \in \Delta_E$, Eq. 2 can simplified as $p_E/\Delta = (1-\epsilon)/\Delta + \epsilon p_E^{\mu}/\Delta$, where $p_E^{\mu} = \widehat{\mu}(x \in \Delta_E)$. Thus, $p_E^{\mu} = \frac{p_E}{\epsilon} - \frac{1-\epsilon}{\epsilon} \geq 0$ if $\epsilon \geq 1 - p_E$. Obviously, $p_E^{\mu} = \frac{p_E}{\epsilon} - \frac{1-\epsilon}{\epsilon} \leq \frac{1}{\epsilon} - \frac{1-\epsilon}{\epsilon} = 1$ guaranteed by the validity of $\widehat{p}^{s,a}$. (2) When $x \notin \Delta_E$, we have $p_i/\Delta = \epsilon p_i^{\mu}/\Delta$, i.e.,When $x \notin \Delta_E$, We immediately have $p_i^{\mu} = \frac{p_i}{\epsilon} \leq \frac{1-p_E}{\epsilon} \leq 1$ when $\epsilon \geq 1 - p_E$. Also, $p_i^{\mu} = \frac{p_i}{\epsilon} \geq 0$.

**Sufficiency.** (1) When $x \in \Delta_E$, let $p_E^{\mu} = \frac{p_E}{\epsilon} - \frac{1-\epsilon}{\epsilon} \geq 0$, we have $\epsilon \geq 1 - p_E$. $p_E^{\mu} = \frac{p_E}{\epsilon} - \frac{1-\epsilon}{\epsilon} \leq 1$ in nature. (2) When $x \notin \Delta_E$, $p_i^{\mu} = \frac{p_i}{\epsilon} \geq 0$ in nature. Let $p_i^{\mu} = \frac{p_i}{\epsilon} \leq 1$, we have $p_i \leq \epsilon$. We need to take the intersection set of (1) and (2), and we find that $\epsilon \geq 1 - p_E \Rightarrow \epsilon \geq 1 - p_E \geq p_i$ that satisfies the condition in (2). Thus, the intersection set of (1) and (2) would be $\epsilon \geq 1 - p_E$.

In summary, as $\epsilon \geq 1 - p_E$ is both the necessary and sufficient condition, we have the conclusion that $\widehat{\mu}(x)$ is a valid probability density function $\iff \epsilon \geq 1 - p_E$.

$\square$

## C  EQUIVALENCE BETWEEN CATEGORICAL REPRESENTATION AND HISTOGRAM ESTIMATION IN DISTRIBUTIONAL RL

**Proposition 5.** *Suppose the target categorical distribution $c = \sum_{i=1}^{N} p_i \delta_{z_i}$ and the target histogram function $h(x) = \sum_{i=1}^{N} p_i \mathbb{1}(x \in \Delta_i)/\Delta$, updating the parameterized categorical distribution $c_{\theta}$ under KL divergence is equivalent to updating the parameterized histogram function $h_{\theta}$.*

*Proof.* For the histogram density estimator $h_{\theta}$ and the true target density function $p(x)$, we can simplify the KL divergence as follows.

$$
\begin{aligned}
D_{\mathrm{KL}}(h, h_{\theta}) &= \sum_{i=1}^{N} \int_{l_{i-1}}^{l_i} \frac{p_i(x)}{\Delta} \log \frac{\frac{p_i(x)}{\Delta}}{\frac{h_{\theta}^i}{\Delta}} dx \\
&= \sum_{i=1}^{N} \int_{l_{i-1}}^{l_i} \frac{p_i(x)}{\Delta} \log \frac{p_i(x)}{\Delta} dx - \sum_{i=1}^{N} \int_{l_{i-1}}^{l_i} \frac{p_i(x)}{\Delta} \log \frac{h_{\theta}^i}{\Delta} dx \\
&\overset{(a)}{\propto} - \sum_{i=1}^{N} \int_{l_{i-1}}^{l_i} \frac{p_i(x)}{\Delta} \log \frac{h_{\theta}^i}{\Delta} dx \\
&\overset{(b)}{=} - \sum_{i=1}^{N} p_i \log \frac{h_{\theta}^i}{\Delta} \\
&\overset{(c)}{\propto} - \sum_{i=1}^{N} p_i \log h_{\theta}^i
\end{aligned}
\tag{11}
$$

where $h_\theta^i$ is determined by $i$ and $\theta$, which is independent of $x$. $(a)$ is true because the target distribution with all $p_i$ is fixed. $(b)$ follows because $p_i(x)$ remains constant for $x \in [l_i, l_{i+1}]$. Finally, $(c)$ holds as the remaining term involving $p_i$ and $\Delta$ is also constant.

On the other hand, we consider the KL-based objective function in learning categorical distribution estimator. Given the target categorical distribution $c = \sum_{i=1}^N p_i \delta_{z_i}$, where the probability $p_i$ is fixed for each atom $z_i$, we aim at updating the current categorical estimator $c_\theta$. Then, we have:

$$D_{\text{KL}}(c, c_\theta) = \sum_{i=1}^N p_i \log \frac{p_i}{c_\theta^i} = \sum_{i=1}^N p_i \log p_i - \sum_{i=1}^N p_i \log c_\theta^i \propto - \sum_{i=1}^N p_i \log c_\theta^i, \qquad (12)$$

where $c_\theta = \sum_{i=1}^N c_\theta^i \delta_{z_i}$ is the current categorical estimator and $c_\theta^i$ is the learnable probability. By comparing the final loss function forms in Eq. 11 and Eq. 12, it turns out that they are equivalent as both $c_\theta^i$ and $h_\theta^i$ are the learnable probabilities, which are parameterized by the same neural network.

$\square$

**Remark.** In CDRL, we use a discrete categorical distribution with probabilities centered on the fixed atoms $\{z_i\}_{i=1}^N$. In contrast, the histogram density estimator in our analysis is a continuous function defined on $[z_0, z_N]$, enabling more nuanced analysis within continuous functions. Proposition 5 indicates that minimizing the KL divergence with the categorical distribution in Eq. 12 amounts to the cross-entropy loss with the parameterized histogram function in Eq. 11.

# D  CONVERGENCE GUARANTEE OF HISTOGRAM DENSITY ESTIMATOR IN DISTRIBUTIONAL RL

**Histogram Function Parameterization Error: Uniform Convergence in Probability.** The previous discrete categorical parameterization error bound in (Rowland et al., 2018) (Proposition 3) is derived between the true return distribution and the limiting return distribution denoted as $\eta_C$ iteratively updated via the Bellman operator $\Pi_C \mathfrak{T}^\pi$ *in expectation*, without considering an asymptotic analysis when the number of sampled $\{s_i, a_i\}_{i=1}^n$ pairs goes to infinity. As a complementary result, we provide a uniform convergence rate for the histogram density estimator in the context of distributional RL. In this particular analysis within this subsection, we denote $\widehat{p}_C^{s,a}$ as the density function estimator for the true limiting return distribution $\eta_C$ via $\Pi_C \mathfrak{T}^\pi$ with its true density $p_C^{s,a}$. In Theorem 2, we show that the sample-based histogram estimator $\widehat{p}_C^{s,a}$ can approximate any arbitrary continuous limiting density function $p_C^{s,a}$ under a mild condition. ***This ensures the use of a histogram density estimator in the implementation of our subsequent algorithm adapted from CDRL.***

**Theorem 2.** *(Uniform Convergence Rate in Probability) Suppose $p_C^{s,a}(x)$ is Lipschitz continuous and the support of a random variable is partitioned by N bins with bin size $\Delta$. Then*

$$\sup_x |\widehat{p}_C^{s,a}(x) - p_C^{s,a}(x)| = O(\Delta) + O_P\left(\sqrt{\frac{\log N}{n\Delta^2}}\right). \qquad (13)$$

*Proof.* Our proof is mainly based on the non-parametric statistics analysis (Wasserman, 2006). In particular, the difference of $\widehat{p}_C^{s,a}(x) - p_C^{s,a}(x)$ can be written as

$$\widehat{p}_C^{s,a}(x) - p_C^{s,a}(x) = \underbrace{\mathbb{E}\left(\widehat{p}_C^{s,a}(x)\right) - p_C^{s,a}(x)}_{\text{bias}} + \underbrace{\widehat{p}_C^{s,a}(x) - \mathbb{E}\left(\widehat{p}_C^{s,a}(x)\right)}_{\text{stochastic variation}}. \qquad (14)$$

**(1) The first bias term.** Without loss of generality, we consider $x \in \Delta_k$, we have

$$\begin{aligned}
\mathbb{E}\left(\widehat{p}_C^{s,a}(x)\right) &= \frac{P(X \in \Delta_k)}{\Delta} \\
&= \frac{\int_{l_0+(k-1)\Delta}^{l_0+k\Delta} p(y)dy}{\Delta} \\
&= \frac{F(l_0 + (k-1)\Delta) - F(l_0 + (k-1)\Delta)}{l_0 + k\Delta - (l_0 + (k-1)\Delta)} \\
&= p_C^{s,a}(x'),
\end{aligned} \qquad (15)$$

where the last equality is based on the mean value theorem. According to the L-Lipschitz continuity property, we have

$$|\mathbb{E}\left(\widehat{p}_{\mathcal{C}}^{s,a}(x)\right) - p_{\mathcal{C}}^{s,a}(x)| = |p_{\mathcal{C}}^{s,a}(x') - p_{\mathcal{C}}^{s,a}(x)| \leq L|x' - x| \leq L\Delta \tag{16}$$

**(2) The second stochastic variation term.** If we let $x \in \Delta_k$, then $\widehat{p}_{\mathcal{C}}^{s,a} = p_k = \frac{1}{n}\sum_{i=1}^{n}\mathbb{1}(X_i \in \Delta_k)$, we thus have

$$
\begin{aligned}
&P\left(\sup_x |\widehat{p}_{\mathcal{C}}^{s,a}(x) - \mathbb{E}\left(\widehat{p}_{\mathcal{C}}^{s,a}(x)\right)| > \epsilon\right) \\
&= P\left(\max_{j=1,\cdots,N} \left|\frac{1}{n}\sum_{i=1}^{n}\mathbb{1}\left(X_i \in \Delta_j\right)/\Delta - P\left(X_i \in \Delta_j\right)/\Delta\right| > \epsilon\right) \\
&= P\left(\max_{j=1,\cdots,N} \left|\frac{1}{n}\sum_{i=1}^{n}\mathbb{1}\left(X_i \in \Delta_j\right) - P\left(X_i \in \Delta_j\right)\right| > \Delta\epsilon\right) \\
&\leq \sum_{j=1}^{N} P\left(\left|\frac{1}{n}\sum_{i=1}^{n}\mathbb{1}\left(X_i \in \Delta_j\right) - P\left(X_i \in \Delta_j\right)\right| > \Delta\epsilon\right) \\
&\leq N \cdot \exp\left(-2n\Delta^2\epsilon^2\right) \quad \text{(by Hoeffding's inequality),}
\end{aligned}
\tag{17}
$$

where in the last inequality we know that the indicator function is bounded in [0, 1]. We then let the last term be a constant independent of $N, n, \Delta$ and simplify the order of $\epsilon$. Then, we have:

$$\sup_x |\widehat{p}_{\mathcal{C}}^{s,a}(x) - \mathbb{E}\left(\widehat{p}_{\mathcal{C}}^{s,a}(x)\right)| = O_P\left(\sqrt{\frac{\log N}{n\Delta^2}}\right) \tag{18}$$

In summary, as the above inequality holds for each $x$, we thus have the uniform convergence rate of a histogram density estimator

$$
\begin{aligned}
\sup_x |\widehat{p}_{\mathcal{C}}^{s,a}(x) - p_{\mathcal{C}}^{s,a}(x)| &\leq \sup_x |\mathbb{E}\left(\widehat{p}_{\mathcal{C}}^{s,a}(x)\right) - p_{\mathcal{C}}^{s,a}(x)| + \sup_x |\widehat{p}_{\mathcal{C}}^{s,a}(x) - \mathbb{E}\left(\widehat{p}_{\mathcal{C}}^{s,a}(x)\right)| \\
&= O\left(\Delta\right) + O_P\left(\sqrt{\frac{\log N}{n\Delta^2}}\right).
\end{aligned}
\tag{19}
$$

$\square$

# E DISCUSSION ABOUT KL DIVERGENCE IN DISTRIBUTIONAL RL

## E.1 PROPERTIES OF KL DIVERGENCE IN DISTRIBUTIONAL RL

**Remark on KL Divergence.** As stated in Section 3 of CDRL (Bellemare et al., 2017a), when the categorical parameterization is applied after the projection operator $\Pi_{\mathcal{C}}$, the distributional Bellman operator $\mathfrak{T}^\pi$ has the contraction guarantee under Cramér distance or Wasserstein distance (Rowland et al., 2018), albeit the direct use of a non-expansive KL divergence (Morimura et al., 2011). Similarly, our histogram density parameterization with the projection $\Pi_{\mathcal{C}}$ and KL divergence also enjoys a contraction property due to the equivalence between optimizing histogram function and categorical distribution analyzed in Appendix C. We summarize some properties of KL divergence in distributional RL in Proposition 6.

**Proposition 6.** *Given two probability measures $\mu$ and $\nu$, we define the supreme $D_{KL}$ as a functional $\mathcal{P}(\mathcal{X})^{\mathcal{S}\times\mathcal{A}} \times \mathcal{P}(\mathcal{X})^{\mathcal{S}\times\mathcal{A}} \to \mathbb{R}$, i.e., $D_{KL}^\infty(\mu, \nu) = \sup_{(s,a)\in\mathcal{S}\times\mathcal{A}} D_{KL}(\mu(s,a), \nu(s,a))$. we have:*

*(1) $\mathfrak{T}^\pi$ is a non-expansive distributional Bellman operator under $D_{KL}^\infty$, i.e.,*

$$D_{KL}^\infty(\mathfrak{T}^\pi Z_1, \mathfrak{T}^\pi Z_2) \leq D_{KL}^\infty(Z_1, Z_2), \tag{20}$$

*(2) $D_{KL}^\infty(Z_n, Z) \to 0$ implies the Wasserstein distance $W_p(Z_n, Z) \to 0$.*

*Proof.* We first assume $Z_\theta$ is absolutely continuous and the supports of two distributions in KL divergence have a negligible intersection (Arjovsky & Bottou, 2017), under which the KL divergence is well-defined.

(1) The contraction analysis of distributional Bellman operator $\mathfrak{T}^\pi$ under a distribution divergence $d_p$ depends on its *scale sensitive* (**S**) and *sum invariant* (**I**) properties (Bellemare et al., 2017b;a). We say $d_p$ is scale sensitive (of order $\tau$) if there exists a $\tau > 0$, such that for all random variables $X, Y$ and a real value $a > 0$, $d_p(aX, aY) \leq |a|^\tau d_p(X, Y)$. $d_p$ has the sum invariant property if whenever a random variable $A$ is independent from $X, Y$, we have $d_p(A + X, A + Y) \leq d_p(X, Y)$. We first prove that the $D_{KL}$ is sum-invariant, which is based on the dual form of KL divergence via the variational representation (Donsker & Varadhan, 1976; Agrawal & Horel, 2021):

$$D_{\mathrm{KL}}(X, Y) = \sup_{f \in \mathcal{L}^b} \{\mathbb{E}_X[f(x)] - \log \left( \mathbb{E}_Y \left[ e^{f(y)} \right] \right)\}, \tag{21}$$

where $\mathcal{L}^b$ is the space of bounded measurable functions. Consequently, we have

$$
\begin{aligned}
D_{\mathrm{KL}}(A + X, A + Y) &= \sup_{f \in \mathcal{L}^b} \{\mathbb{E}_{Z_1 = A + X}[f(z_1)] - \log \left( \mathbb{E}_{Z_2 = A + Y} \left[ e^{f(z_2)} \right] \right)\} \\
&\overset{(a)}{=} \sup_{f \in \mathcal{L}^b} \{\mathbb{E}_A \left[ \mathbb{E}_X \left[ f(x + a) \right] \right] - \log \left( \mathbb{E}_A \left[ \mathbb{E}_Y \left[ e^{f(y+a)} \right] \right] \right)\} \\
&\overset{(b)}{\leq} \sup_{f \in \mathcal{L}^b} \{\mathbb{E}_A \mathbb{E}_X[f(x + a)] - \mathbb{E}_A \log \left( \mathbb{E}_Y \left[ e^{f(y+a)} \right] \right)\} \\
&= \sup_{f \in \mathcal{L}^b} \{\mathbb{E}_A[\mathbb{E}_X[f(x + a)] - \log \left( \mathbb{E}_Y \left[ e^{f(y+a)} \right] \right)]\} \\
&\overset{(c)}{\leq} \mathbb{E}_A \sup_{f \in \mathcal{L}^b} \{\mathbb{E}_X[f(x + a)] - \log \left( \mathbb{E}_Y \left[ e^{f(y+a)} \right] \right)\} \\
&\overset{(d)}{=} \mathbb{E}_A \sup_{g \in \mathcal{L}^b} \{\mathbb{E}_X[g(x)] - \log \left( \mathbb{E}_Y \left[ e^{g(y)} \right] \right)\} \\
&= D_{\mathrm{KL}}(X, Y),
\end{aligned}
\tag{22}
$$

where (a) results from the independence between $A$ and $X$ ($Y$). (b) and (c) rely on the Jensen inequality for the function $-\log$ and the operator $\sup$. (d) is because the translation is still within the same bounded functional space. Next, we show that $D_{\mathrm{KL}}$ is not scale-sensitive, where we denote the probability density function of $X$ and $Y$ as $p$ and $q$.

$$D_{\mathrm{KL}}(aX, aY) = \int_{-\infty}^\infty \frac{1}{a} p \left( \frac{x}{a} \right) \log \frac{\frac{1}{a} p \left( \frac{x}{a} \right)}{\frac{1}{a} q \left( \frac{x}{a} \right)} \mathrm{d}x = \int_{-\infty}^\infty p(y) \log \frac{p(y)}{q(y)} \mathrm{d}y = D_{\mathrm{KL}}(X, Y) \tag{23}$$

Putting the two properties together and given two return distributions $Z_1(s, a)$ and $Z_2(s, a)$, we have the non-expansive contraction property of the supremal form of $D_{\mathrm{KL}}$ as follows.

$$
\begin{aligned}
D_{\mathrm{KL}}^\infty(\mathfrak{T}^\pi Z_1, \mathfrak{T}^\pi Z_2) &= \sup_{s,a} D_{\mathrm{KL}}(\mathfrak{T}^\pi Z_1(s, a), \mathfrak{T}^\pi Z_2(s, a)) \\
&= \sup_{s,a} D_{\mathrm{KL}}(R(s, a) + \gamma Z_1(s', a'), R(s, a) + \gamma Z_2(s', a')) \\
&\overset{(a)}{\leq} D_{\mathrm{KL}}(\gamma Z_1(s', a'), \gamma Z_2(s', a')) \\
&\overset{(b)}{=} D_{\mathrm{KL}}(Z_1(s', a'), Z_2(s', a')) \\
&\leq \sup_{s,a} D_{\mathrm{KL}}(Z_1(s', a'), Z_2(s', a')) \\
&= D_{\mathrm{KL}}^\infty(Z_1, Z_2),
\end{aligned}
\tag{24}
$$

where (a) relies on the sum invariant property of $D_{\mathrm{KL}}$ and (b) utilizes the non-scale sensitive property of $D_{\mathrm{KL}}$. By applying the well-known Banach fixed point theorem, we have a unique return distribution when convergence of distributional dynamic programming under $D_{\mathrm{KL}}^\infty$.

(2) By the definition of $D_{\mathrm{KL}}^{\infty}$, we have $\sup_{s,a} D_{\mathrm{KL}}(Z_n(s,a), Z(s,a)) \to 0$ implies $D_{\mathrm{KL}}(Z_n, Z) \to 0$. $D_{\mathrm{KL}}(Z_n, Z) \to 0$ implies the total variation distance $\delta(Z_n, Z) \to 0$ according to a straightforward application of Pinsker's inequality

$$\delta\left(Z_n, Z\right) \le \sqrt{\frac{1}{2} D_{\mathrm{KL}}\left(Z_n, Z\right)} \to 0, \quad \delta\left(Z, Z_n\right) \le \sqrt{\frac{1}{2} D_{\mathrm{KL}}\left(Z, Z_n\right)} \to 0 \qquad (25)$$

Based on Theorem 2 in WGAN (Arjovsky et al., 2017), $\delta(Z_n, Z) \to 0$ implies $W_p(Z_n, Z) \to 0$. This is trivial by recalling the fact that $\delta$ and $W$ give the strong and weak topologies on the dual of $(C(\mathcal{X}), \|\cdot\|_\infty)$ when restricted to $\mathrm{Prob}(\mathcal{X})$.

$\square$

### E.2 Equivalence between Cross-Entropy Loss and KL Divergence in Neural FZI

If the target density function in evaluating the KL divergence is not fixed, using cross-entropy loss instead of the KL divergence may underestimate the uncertainty of return since this simplification may fail to capture the exact shape or uncertainty spread of the true target return distribution. However, this underestimation issue does occur in our analysis. Particularly, the leverage of target network in Neural FZI, which is fixed in the updating of each phase, guarantees that the KL divergence is *exactly* proportional to the cross-entropy loss. Figure 6 suggests that C51 with cross-entropy loss (DSAC_CE) behaves similarly to the vanilla C51 equipped with KL divergence (DSAC) in both three Atari games and MuJoCo environments with continuous action space.

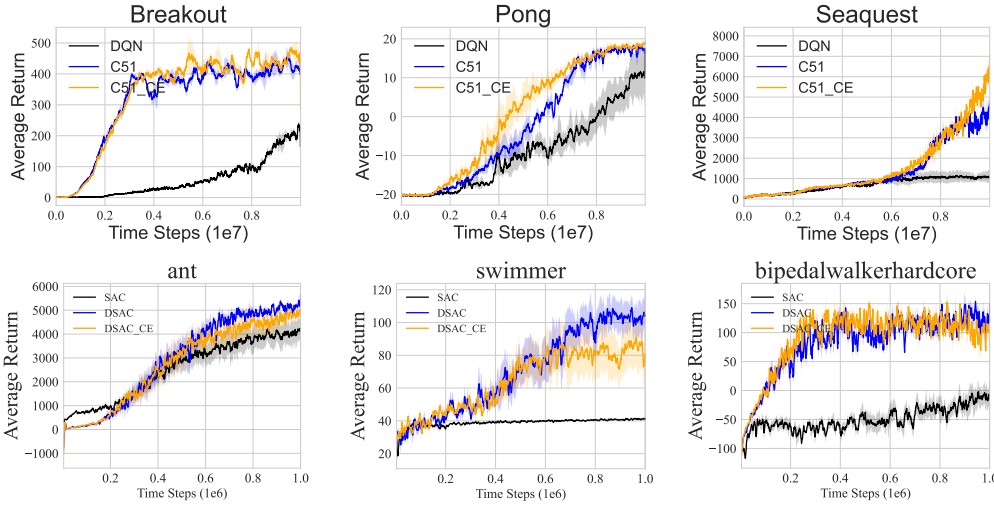

Figure 6: (**First row**) Learning curves of C51 under cross-entropy loss on Atari games over 3 seeds. (**Second row**) Learning curves of DSAC with C51 under cross-entropy loss on MuJoCo environments over 5 seeds.

## F   Proof of Proposition 2

**Proposition 2** (Decomposed Neural FZI) Denote $q_\theta^{s,a}$ as the histogram density function of $Z_\theta^k(s,a)$ in Neural FZI. Based on Eq. 2 and KL divergence as $d_p$, Neural FZI in Eq. 1 is simplified as

$$Z_\theta^{k+1} = \operatorname*{argmin}_{q_\theta} \frac{1}{n} \sum_{i=1}^{n} [\underbrace{-\log q_\theta^{s_i, a_i}(\Delta_E^i)}_{(a)} + \alpha \mathcal{H}(\widehat{\mu}^{s_i', \pi_Z(s_i')}, q_\theta^{s_i, a_i})]. \qquad (26)$$

*Proof.* Firstly, given a fixed $p(x)$ we know that minimizing $D_{KL}(p, q_\theta)$ is equivalent to minimizing $\mathcal{H}(p, q)$ by following

$$
\begin{aligned}
D_{KL}(p, q_\theta) &= \sum_{i=1}^N \int_{l_{i-1}}^{l_i} \frac{p_i(x)}{\Delta} \log \frac{p^i(x)/\Delta}{q_\theta^i/\Delta} \, dx \\
&= -\sum_{i=1}^N \int_{l_{i-1}}^{l_i} \frac{p_i(x)}{\Delta} \log \frac{q_\theta^i}{\Delta} \, dx - \left( \sum_{i=1}^N \int_{l_{i-1}}^{l_i} \frac{p_i(x)}{\Delta} \log \frac{p^i(x)}{\Delta} \, dx \right) \quad (27) \\
&= \mathcal{H}(p, q_\theta) - \mathcal{H}(p) \\
&\propto \mathcal{H}(p, q_\theta)
\end{aligned}
$$

where $p = \sum_{i=1}^N p_i(x) \mathbb{1}(x \in \Delta^i)/\Delta$ and $q_\theta = \sum_{i=1}^N q_i/\Delta$. Based on $\mathcal{H}(p, q_\theta)$, we use $p^{s_i', \pi_Z(s_i')}(x)$ to denote the target probability density function of the random variable $\mathcal{R}(s_i, a_i) + \gamma Z_{\theta^*}^k (s_i', \pi_Z(s_i'))$. Then, we can derive the objective function within each Neural FZI as

$$
\frac{1}{n} \sum_{i=1}^n \mathcal{H}(p^{s_i', \pi_Z(s_i')}, q_\theta^{s_i, a_i})
$$

$$
= \frac{1}{n} \sum_{i=1}^n \left( -(1-\epsilon) \sum_{j=1}^N \int_{l_{j-1}}^{l_j} \frac{\mathbb{1}(x \in \Delta_E^i)}{\Delta} \log \frac{q_\theta^{s_i, a_i}(\Delta_j)}{\Delta} dx - \epsilon \sum_{j=1}^N \int_{l_{j-1}}^{l_j} \frac{p_j^\mu}{\Delta} \log \frac{q_\theta^{s_i, a_i}(\Delta_j)}{\Delta} dx \right)
$$

$$
= \frac{1}{n} \sum_{i=1}^n \left( (1-\epsilon)(-\log q_\theta^{s_i, a_i}(\Delta_E^i)) + \epsilon \mathcal{H}(\widehat{\mu}^{s_i', \pi_Z(s_i')}, q_\theta^{s_i, a_i}) \right) + (1-\epsilon)\Delta
$$

$$
\propto \frac{1}{n} \sum_{i=1}^n \left( -\log q_\theta^{s_i, a_i}(\Delta_E^i) + \alpha \mathcal{H}(\widehat{\mu}^{s_i', \pi_Z(s_i')}, q_\theta^{s_i, a_i}) \right), \text{ where } \alpha = \frac{\epsilon}{1-\epsilon} > 0
$$

$$(28)$$

where recall that $\widehat{\mu}^{s_i', \pi_Z(s_i')} = \sum_{i=1}^N p_i^\mu(x) \mathbb{1}(x \in \Delta_i)/\Delta = \sum_{i=1}^N p_i^\mu/\Delta$ for conciseness and denote $q_\theta^{s_i, a_i} = \sum_{j=1}^N q_\theta^{s_i, a_i}(\Delta_j)/\Delta$. The cross-entropy $\mathcal{H}(\widehat{\mu}^{s_i', \pi_Z(s_i')}, q_\theta^{s_i, a_i})$ is based on the discrete distribution when $i = 1, ..., N$. $\Delta_E^i$ represent the interval that $\mathbb{E}\left[ \mathcal{R}(s_i, a_i) + \gamma Z_{\theta^*}^k (s_i', \pi_Z(s_i')) \right]$ falls into, i.e., $\mathbb{E}\left[ \mathcal{R}(s_i, a_i) + \gamma Z_{\theta^*}^k (s_i', \pi_Z(s_i')) \right] \in \Delta_E^i$. $\qquad\square$

# G  PROOF OF PROPOSITION 3

**Proposition 3** (Equivalence between **the term (a)** in Decomposed Neural FZI and Neural FQI) In Eq. 3 of Neural FZI, assume the function class $\{Z_\theta : \theta \in \Theta\}$ is sufficiently large such that it contains the target $\{Y_i^k\}_{i=1}^n$, when $\Delta \to 0$, for all $k$, minimizing **the term (a)** in Eq. 3 implies

$$
P(Z_\theta^{k+1}(s, a) = \mathcal{T}^{opt} Q_{\theta^*}^k(s, a)) = 1, \quad \text{and} \quad \int_{-\infty}^{+\infty} \left| F_{q_\theta}(x) - F_{\delta_{\mathcal{T}^{opt} Q_{\theta^*}^k(s,a)}}(x) \right| dx = o(\Delta),
$$

$$(29)$$

where $\mathcal{T}^{opt} Q_{\theta^*}^k(s, a)$ is the scalar-valued target in the k-th phase of Neural FQI, and $\delta_{\mathcal{T}^{opt} Q_{\theta^*}^k(s,a)}$ is the Dirac delta function defined on the scalar $\mathcal{T}^{opt} Q_{\theta^*}^k(s, a)$.

*Proof.* **Limiting Case.** Firstly, we define the distributional Bellman optimality operator $\mathfrak{T}^{opt}$ as follows:

$$
\mathfrak{T}^{opt} Z(s, a) \overset{D}{=} \mathcal{R}(s, a) + \gamma Z(S', a^*), \quad (30)
$$

where $S' \sim P(\cdot \mid s, a)$ and $a^* = \underset{a'}{\mathrm{argmax}} \mathbb{E}[Z(S', a')]$. If $\{Z_\theta : \theta \in \Theta\}$ is sufficiently large enough such that it contains $\mathfrak{T}^{opt} Z_{\theta^*} (\{Y_i^k\}_{i=1}^n)$, then optimizing Neural FZI in Eq. 1 leads to $Z_\theta^{k+1} = \mathfrak{T}^{opt} Z_{\theta^*}$.

Secondly, we apply the return density decomposition on the target histogram function $\widehat{p}^{s,a}(x)$. Consider the parameterized histogram density function $h_\theta$ and denote $h_\theta^E/\Delta$ as the bin height in the

bin $\Delta_E$, under the KL divergence between the first histogram function $\mathbb{1}(x \in \Delta_E)$ with $h_\theta(x)$, the objective function is simplified as

$$D_{\text{KL}}(\mathbb{1}(x \in \Delta_E)/\Delta, h_\theta(x)) = -\int_{x \in \Delta_E} \frac{1}{\Delta} \log \frac{\frac{h_\theta^E}{\Delta}}{\frac{1}{\Delta}} dx = -\log h_\theta^E \tag{31}$$

Since $\{Z_\theta : \theta \in \Theta\}$ is sufficiently large enough that can represent the pdf of $\{Y_i^k\}_{i=1}^n$, it also implies that $\{Z_\theta : \theta \in \Theta\}$ can represent the term (a) part in its pdf via the return density decomposition. The KL minimizer would be $\widehat{h}_\theta = \mathbb{1}(x \in \Delta_E)/\Delta$ in expectation. Then, $\lim_{\Delta \to 0} \arg\min_{h_\theta} D_{\text{KL}}(\mathbb{1}(x \in \Delta_E)/\Delta, h_\theta(x)) = \delta_{\mathbb{E}[Z^{\text{target}}(s,a)]}$, where $\delta_{\mathbb{E}[Z^{\text{target}}(s,a)]}$ is a Dirac Delta function centered at $\mathbb{E}\left[Z^{\text{target}}(s,a)\right]$ and can be viewed as a generalized probability density function. That being said, the limiting probability density function (pdf) converges to a Dirac delta function at $\mathbb{E}\left[Z^{\text{target}}(s,a)\right]$. The limit behavior from a histogram function $\widehat{p}$ to a continuous one for $Z^{\text{target}}$ is guaranteed by Theorem 2, and this also applies from $h_\theta$ to $Z_\theta$. In Neural FZI, we have $Z^{\text{target}} = \mathfrak{T}^{\text{opt}} Z_{\theta^*}$. Here we use $Z_\theta^{k+1}(s,a)$ as the random variable whose cdf is the limiting distribution. According to the definition of the Dirac function, in the limiting case where $\Delta \to 0$, we attain that

$$\mathbb{P}(Z_\theta^{k+1}(s,a) = \mathbb{E}\left[\mathfrak{T}^{\text{opt}} Z_{\theta^*}^k(s,a)\right]) = 1. \tag{32}$$

This is because the pdf of the limiting return random variable $Z_\theta^{k+1}(s,a)$ is a Dirac delta function, which implies that the random variable takes this constant value with probability one. Due to the linearity of expectation in Lemma 4 of (Bellemare et al., 2017a), we have

$$\mathbb{E}\left[\mathfrak{T}^{\text{opt}} Z_{\theta^*}^k(s,a)\right] = \mathfrak{T}^{\text{opt}} \mathbb{E}\left[Z_{\theta^*}^k(s,a)\right] = \mathcal{T}^{\text{opt}} Q_{\theta^*}^k(s,a) \tag{33}$$

Finally, we obtain the convergence in probability one in the limiting case:

$$\mathbb{P}(Z_\theta^{k+1}(s,a) = \mathcal{T}^{\text{opt}} Q_{\theta^*}^k(s,a)) = 1 \quad \text{as } \Delta \to 0 \tag{34}$$

**Convergence in Distribution.** The connection established above is in the limiting case. Alternatively, we can have a more formal proof by using the language of convergence in distribution. Here, we use $Z_{\theta,\Delta}^{k+1}$ to replace $Z_\theta^{k+1}$ to explicitly consider its asymptotic behavior. According to the fact that $\infty\{x \in \Delta_E\}/\Delta$ is the optimizer when minimizing the term (a) in Eq. 3 given a fixed $\Delta$, the convergence in distribution is:

$$\lim_{\Delta \to 0} \mathcal{D}(Z_{\theta,\Delta}^{k+1}) = \lim_{\Delta \to 0} \mathcal{D}(\mathbb{1}\{x \in \Delta_E\}/\Delta) = \mathcal{D}(\delta_{\mathcal{T}^{\text{opt}} Q_{\theta^*}^k(s,a)}), \tag{35}$$

where $\delta_{\mathcal{T}^{\text{opt}} Q_{\theta^*}^k(s,a)}$ is the Dirac Delta function centered at $\mathcal{T}^{\text{opt}} Q_{\theta^*}^k(s,a)$. $\mathcal{D}(\delta_{\mathcal{T}^{\text{opt}} Q_{\theta^*}^k(s,a)})$ is the corresponding step function, where $\mathcal{D}(\delta_{\mathcal{T}^{\text{opt}} Q_{\theta^*}^k(s,a)})(x) = 1$ if $x \geq \mathcal{T}^{\text{opt}} Q_{\theta^*}^k(s,a)$, and equals 0 otherwise. Note that the convergence in distribution in terms of the Dirac delta function implies that $\mathbb{P}(Z_\theta^{k+1}(s,a) = \mathcal{T}^{\text{opt}} Q_{\theta^*}^k(s,a)) = 1$ as $\Delta \to 0$ in Eq 34.

**Convergence Rate.** In order to characterize how the difference varies when $\Delta \to 0$, we further define $\Delta_E = [l_e, l_{e+1})$ and we have:

$$\int_{-\infty}^{+\infty} \left| F_{q_\theta}(x) - F_{\delta_{\mathcal{T}^{\text{opt}} Q_{\theta^*}^k(s,a)}}(x) \right| dx = \frac{1}{2\Delta} \left( \left(\mathcal{T}^{\text{opt}} Q_{\theta^*}^k(s,a) - l_e\right)^2 + \left(l_{e+1} - \mathcal{T}^{\text{opt}} Q_{\theta^*}^k(s,a)\right)^2 \right)$$

$$= \frac{1}{2\Delta}(a^2 + (\Delta - a)^2)$$

$$\leq \Delta/2$$

$$= o(\Delta), \tag{36}$$

where $\mathcal{T}^{\text{opt}} Q_{\theta^*}^k(s,a) = \mathbb{E}\left[\mathfrak{T}^{\text{opt}} Z_{\theta^*}^k(s,a)\right] \in \Delta_E$ and we denote $a = \mathcal{T}^{\text{opt}} Q_{\theta^*}^k(s,a) - l_e$. The first equality holds as $q_\theta(x)$, the KL minimizer while minimizing the term (a), would follows a uniform distribution on $\Delta_E$, i.e., $\widehat{q}_\theta = \mathbb{1}(x \in \Delta_E)/\Delta$. Thus, the integral of LHS would be the area of two centralized triangles accordingly. The inequality is because the maximizer is obtained when $a = \Delta$ or 0. The result implies that the convergence rate in distribution difference is $o(\Delta)$.

$\square$

## H  CONVERGENCE PROOF OF DERPI IN THEOREM 1

### H.1  PROOF OF DISTRIBUTION-ENTROPY-REGULARIZED POLICY EVALUATION IN LEMMA 1

**Lemma 1**(Distribution-Entropy-Regularized Policy Evaluation) Consider the distribution-entropy-regularized Bellman operator $\mathcal{T}_d^\pi$ in Eq. 6 and assume $\mathcal{H}(\mu^{s_t,a_t}, q_\theta^{s_t,a_t})$ is bounded for all $(s_t, a_t) \in \mathcal{S} \times \mathcal{A}$. Define $Q^{k+1} = \mathcal{T}_d^\pi Q^k$, then $Q^{k+1}$ will converge to a *corrected* Q-value of $\pi$ as $k \to \infty$ with the new objective function $J'(\pi)$ defined as

$$J'(\pi) = \sum_{t=0}^{T} \mathbb{E}_{(s_t,a_t) \sim \rho_\pi} \left[ r(s_t, a_t) + \gamma f(\mathcal{H}(\mu^{s_t,a_t}, q_\theta^{s_t,a_t})) \right]. \tag{37}$$

*Proof.* Firstly, we plug in $V(s_{t+1})$ into RHS of the iteration in Eq. 6, then we obtain

$$\begin{aligned}
&\mathcal{T}_d^\pi Q(s_t, a_t) \\
&= r(s_t, a_t) + \gamma \mathbb{E}_{s_{t+1} \sim P(\cdot|s_t, a_t)} [V(s_{t+1})] \\
&= r(s_t, a_t) + \gamma f(\mathcal{H}(\mu^{s_t,a_t}, q_\theta^{s_t,a_t})) + \gamma \mathbb{E}_{(s_{t+1},a_{t+1}) \sim \rho^\pi} [Q(s_{t+1}, a_{t+1})] \\
&\triangleq r_\pi(s_t, a_t) + \gamma \mathbb{E}_{(s_{t+1},a_{t+1}) \sim \rho^\pi} [Q(s_{t+1}, a_{t+1})],
\end{aligned} \tag{38}$$

where $r_\pi(s_t, a_t) \triangleq r(s_t, a_t) + \gamma f(\mathcal{H}(\mu^{s_t,a_t}, q_\theta^{s_t,a_t}))$ is the entropy augmented reward we redefine. Applying the standard convergence results for policy evaluation (Sutton & Barto, 2018), we can attain that this Bellman updating under $\mathcal{T}_d^\pi$ is convergent under the assumption of $|\mathcal{A}| < \infty$ and bounded entropy augmented rewards $r_\pi$. □

### H.2  POLICY IMPROVEMENT WITH PROOF

**Lemma 2.** *(Distribution-Entropy-Regularized Policy Improvement) Let $\pi \in \Pi$ and a new policy $\pi_{new}$ be updated via the policy improvement step in the policy optimization. Then $Q^{\pi_{new}}(s_t, a_t) \geq Q^{\pi_{old}}(s_t, a_t)$ for all $(s_t, a_t) \in \mathcal{S} \times \mathcal{A}$ with $|\mathcal{A}| \leq \infty$.*

*Proof.* The policy improvement in Lemma 2 implies that $\mathbb{E}_{a_t \sim \pi_{new}}[Q^{\pi_{old}}(s_t, a_t)] \geq \mathbb{E}_{a_t \sim \pi_{old}}[Q^{\pi_{old}}(s_t, a_t)]$, we consider the Bellman equation via the distribution-entropy-regularized Bellman operator $\mathcal{T}_{sd}^\pi$:

$$\begin{aligned}
Q^{\pi_{old}}(s_t, a_t) &\triangleq r(s_t, a_t) + \gamma \mathbb{E}_{s_{t+1} \sim \rho} [V^{\pi_{old}}(s_{t+1})] \\
&= r(s_t, a_t) + \gamma f(\mathcal{H}(\mu^{s_t,a_t}, q_\theta^{s_t,a_t})) + \gamma \mathbb{E}_{(s_{t+1},a_{t+1}) \sim \rho^{\pi_{old}}} [Q^{\pi_{old}}(s_{t+1}, a_{t+1})] \\
&\leq r(s_t, a_t) + \gamma f(\mathcal{H}(\mu^{s_t,a_t}, q_\theta^{s_t,a_t})) + \gamma \mathbb{E}_{(s_{t+1},a_{t+1}) \sim \rho^{\pi_{new}}} [Q^{\pi_{old}}(s_{t+1}, a_{t+1})] \\
&= r_{\pi_{new}}(s_t, a_t) + \gamma \mathbb{E}_{(s_{t+1},a_{t+1}) \sim \rho^{\pi_{new}}} [Q^{\pi_{old}}(s_{t+1}, a_{t+1})] \\
&\quad \vdots \\
&\leq Q^{\pi_{new}}(s_{t+1}, a_{t+1}),
\end{aligned} \tag{39}$$

where we have repeated expanded $Q^{\pi_{old}}$ on the RHS by applying the distribution-entropy-regularized distributional Bellman operator. Convergence to $Q^{\pi_{new}}$ follows from Lemma 1. □

### H.3  PROOF OF DERPI IN THEOREM 1

**Theorem 1** (Distribution-Entropy-Regularized Policy Iteration) Repeatedly applying distribution-entropy-regularized policy evaluation in Eq. 6 and the policy improvement, the policy converges to an optimal policy $\pi^*$ such that $Q^{\pi^*}(s_t, a_t) \geq Q^\pi(s_t, a_t)$ for all $\pi \in \Pi$.

*Proof.* The proof is similar to soft policy iteration (Haarnoja et al., 2018a). For completeness, we provide the proof here. By Lemma 2, as the number of iteration increases, the sequence $Q^{\pi_i}$ at $i$-th iteration is monotonically increasing. Since we assume the uncertainty-aware entropy is bounded, the $Q^\pi$ is thus bounded as the rewards are bounded. Hence, the sequence will converge to some $\pi^*$.

Further, we prove that $\pi^*$ is in fact optimal. At the convergence point, for all $\pi \in \Pi$, it must be case that:

$$\mathbb{E}_{a_t \sim \pi^*}\left[Q^{\pi_{\text{old}}}\left(s_t, a_t\right)\right] \geq \mathbb{E}_{a_t \sim \pi}\left[Q^{\pi_{\text{old}}}\left(s_t, a_t\right)\right].$$

According to the proof in Lemma 2, we can attain $Q^{\pi^*}(s_t, a_t) > Q^\pi(s_t, a_t)$ for $(s_t, a_t)$. That is to say, the "corrected" value function of any other policy in $\Pi$ is lower than the converged policy, indicating that $\pi^*$ is optimal. □

# I  PROOF OF INTERPOLATION FORM OF $\hat{J}_q(\theta)$

In SAC (Haarnoja et al., 2018a) (Section 4.2), it introduces another parameterized state value function to approximate the soft value in the function approximation setting. Instead, we are not intended to do so, but directly use a single Q network to be optimized, which allows the interpolation form of our algorithm. In particular, we directly evaluate the least squared loss between the current Q estimates and the target ones for the critic loss. With a particular form of $f_\pi(\mathcal{H})$, the removal of the interaction term, and the replacement of $Q_\theta$ with $\mathbb{E}\left[q_\theta\right]$, we can derive the interpolation form of $\hat{J}_q(\theta)$ according to the following formula:

$$
\begin{aligned}
\hat{J}_q(\theta) &= \mathbb{E}_{s,a}\left[\left(\mathcal{T}_d^\pi Q_{\theta^*}(s,a) - Q_\theta(s,a)\right)^2\right] \\
&= \mathbb{E}_{s,a}\left[\left(\mathcal{T}^\pi Q_{\theta^*}(s,a) - Q_\theta(s,a) + \gamma(\tau^{1/2}\mathcal{H}^{1/2}(\mu^{s,a}, q_\theta^{s,a})/\gamma)\right)^2\right] \\
&= \mathbb{E}_{s,a}\left[\left(\mathcal{T}^\pi \mathbb{E}\left[q_{\theta^*}(s,a)\right] - \mathbb{E}\left[q_\theta(s,a)\right]\right)^2\right] + \tau\mathbb{E}_{s,a}\left[\mathcal{H}(\mu^{s,a}, q_\theta^{s,a})\right] \\
&\quad + \mathbb{E}_{s,a}\left[\left(\mathcal{T}^\pi \mathbb{E}\left[q_{\theta^*}(s,a)\right] - \mathbb{E}\left[q_\theta(s,a)\right]\right)\mathcal{H}(\mu^{s,a}, q_\theta^{s,a})\right] \\
&\approx \mathbb{E}_{s,a}\left[\left(\mathcal{T}^\pi \mathbb{E}\left[q_{\theta^*}(s,a)\right] - \mathbb{E}\left[q_\theta(s,a)\right]\right)^2\right] + \tau\mathbb{E}_{s,a}\left[\mathcal{H}(\mu^{s,a}, q_\theta^{s,a})\right] \\
&\propto (1-\lambda)\mathbb{E}_{s,a}\left[\left(\mathcal{T}^\pi \mathbb{E}\left[q_{\theta^*}(s,a)\right] - \mathbb{E}\left[q_\theta(s,a)\right]\right)^2\right] + \lambda\mathbb{E}_{s,a}\left[\mathcal{H}(\mu^{s,a}, q_\theta^{s,a})\right],
\end{aligned}
\tag{40}
$$

where the second equation is based on the definition of Distribution-Entropy-Regularized Bellman Operator $\mathcal{T}_d^\pi$ in Eq. 6 and let $f(\mathcal{H}) = (\tau\mathcal{H})^{1/2}/\gamma$. The interaction term $+\mathbb{E}_{s,a}\left[\left(\mathcal{T}^\pi \mathbb{E}\left[q_{\theta^*}(s,a)\right] - \mathbb{E}\left[q_\theta(s,a)\right]\right)\mathcal{H}(\mu^{s,a}, q_\theta^{s,a})\right]$ equal zero in the last equation is rooted in Lemma 1 in (Shi et al., 2022). Although Lemma 1 considers the A/B testing with offline dataset, it demonstrates that the estimation equation between the Bellman error and and any function $\varphi\left(S_t, A_t\right)$ equals zero under mild conditions, such as the consistency assumption. Strictly speaking, we heuristically extend the conclusion in Lemma 1 of (Shi et al., 2022) to the simplification of our critic loss, where we let $\varphi\left(S_t, A_t\right) = \mathcal{H}(\mu^{S_t,A_t}, q_\theta^{S_t,A_t})$. Consequently, we can approximately remove the interaction term as $\mathbb{E}_{s,a}\left[\left(\mathcal{T}^\pi \mathbb{E}\left[q_{\theta^*}(s,a)\right] - \mathbb{E}\left[q_\theta(s,a)\right]\right)\mathcal{H}(\mu^{s,a}, q_\theta^{s,a})\right] = 0$. We set $\lambda = \frac{\tau}{1+\tau} \in [0,1]$. Another simplification is that we directly use $\mathbb{E}\left[q_\theta\right]$ to replace $Q_\theta$ rather than to maintain both two networks $q_\theta$ and $Q_\theta$ with different parameters $\theta$. This strategy simplifies our implementation and contributes to derive the final interpolation form in $\hat{J}_q(\theta)$.

# J  IMPLEMENTATION DETAILS

## J.1  BASELINES ALGORITHMS

**Algorithms in Section 6.1.**

- DQN (Mnih et al., 2015) and C51 (Bellemare et al., 2017a)

- $\mathcal{H}(\mu, q_\theta)(\varepsilon = X)$: a variant of C51 algorithm, where we replace the original target histogram function $\widehat{p}^{s,a}$ with the induced $\widehat{\mu}^{s,a}$ for each $(s,a)$ pair in the update. By varying $\varepsilon = X$, $\mathcal{H}(\mu, q_\theta)$ relies on the distributional loss to different extents in the RL learning. For examples, when $\varepsilon = 1$, $\mathcal{H}(\mu, q_\theta)(\varepsilon = X)$ degenerates to the vanilla C51 algorithm. On the contrary, decreasing $\varepsilon$ in $\mathcal{H}(\mu, q_\theta)$ will reduce the leverage of knowledge from the distributional loss, leading to the performance degradation in distributional learning context.

**Algorithms in Section 6.2.**

- AC: The implementation of AC is directly from the standard SAC algorithm (Haarnoja et al., 2018a) without using the entropy regularization.
- DAC (C51). Based on the original implementation of AC, we employ the C51 loss in the critic loss. Thus, the performance difference between DAC (C51) and AC is merely the leverage of distributional loss.
- DERAC: Our proposed algorithm in Section 5.2 based on the implementation of AC, which uses an interpolated critic loss. The experiments on DERAC are used to validate the convergence analysis in Section 5.2 and highlight the potential performance improvement of an interpolated algorithm in mitigating the over-exploration for an entire distribution RL algorithm.

**Algorithms in Section 6.3.**

- AC: This implementation is same as AC in Section 6.2.
- AC+VE: This is exactly the standard SAC algorithm.
- AC+UE: This implementation is also same as DAC (C51) in Section 6.2, where we use a distributional critic loss in AC algorithm.
- AC+UE+VE: Based on the SAC algorithm, i.e., AC+VE, we additionally use the distribution objective in C51 as the critic loss.

### J.2 REPLACING $\epsilon$ WITH THE RATIO $\varepsilon$ FOR VISUALIZATION

The substitution of $\epsilon$ with $\varepsilon$ is for convenience in the implementation. As Proposition 1 elucidates, the return density decomposition requires that $\epsilon$ exceed certain thresholds to ensure the resultant decomposed $\widehat{\mu}^{s,a}$ qualifies as a valid density function. In practice, pinpointing this lower boundary for $\epsilon$ in each iteration to regulate its range could be prohibitively time-intensive. A more pragmatic approach involves redistributing the mass from the bin that contains the expectation to other bins in specified ratios, thereby introducing the corresponding ratio term $\varepsilon$. By varying $\varepsilon$ from 0 to 1, it invariably meets the validity condition outlined in Proposition 1, thereby streamlining the process for conducting ablation studies concerning $\widehat{\mu}^{s,a}$ as demonstrated in Figure 3.

To delineate the relationship between the ratio $\varepsilon$ and the coefficient $\epsilon$ in constructing $\widehat{\mu}^{s,a}$, after some calculations we establish their equivalence as follows:

$$\varepsilon = \frac{p_E - (1 - \epsilon)}{p_E \epsilon}, \tag{41}$$

where $p_E$ represents the weighting assigned to the bin $\Delta_E$ as specified in Proposition 1. The resulting $\varepsilon \in [0, 1]$ has a monotonically increasing relationship with $\epsilon$, which facilitates the visualization without undermining our conclusion.

**Decomposition Details.** By varying $\varepsilon$, we can evaluate $\epsilon$ via the transformation equation in Eq. 41, which guarantees the validity of return density decomposition. Next, under different $\epsilon$, we compute the induced histogram density $\widehat{\mu}^{s,a}$ via the return density decomposition in Eq. 2. We replace $\widehat{p}^{s,a}$ with $\widehat{\mu}^{s,a}$ in C51 or the critic loss in Distributional AC (C51) in the distributional loss and compare the performance of all considered algorithms. Please refer to the code in the implementation for more details.

### J.3 HYPER-PARAMETERS AND NETWORK STRUCTURE

Our implementation is adapted from the popular RLKit platform. For Distributional SAC with C51, we use 51 atoms similar to the C51 (Bellemare et al., 2017a). For distributional SAC with quantile regression, instead of using fixed quantiles in QR-DQN, we leverage the quantile fraction generation based on IQN (Dabney et al., 2018a) that uniformly samples quantile fractions in order to approximate the full quantile function. In particular, we fix the number of quantile fractions as $N$ and keep them in ascending order. Besides, we adapt the sampling as $\tau_0 = 0, \tau_i = e_i / \sum_{i=0}^{N-1} e_i$, where $\epsilon_i \in U[0, 1], i = 1, ..., N$. We adopt the same hyper-parameters, which are listed in Table 1 and network structure as in the original distributional SAC paper (Ma et al., 2020).

## K DERAC ALGORITHM

We provide a detailed algorithm description of DERAC algorithm in Algorithm 2.

---

**Algorithm 2** Distribution-Entropy-Regularized Actor Critic (DERAC) Algorithm

---

1: Initialize two value networks $q_\theta$, $q_{\theta*}$, and policy network $\pi_\phi$.
2: **for** each iteration **do**
3:    **for** each environment step **do**
4:       $a_t \sim \pi_\phi(a_t|s_t)$.
5:       $s_{t+1} \sim p(s_{t+1}|s_t, a_t)$.
6:       $\mathcal{D} \leftarrow \mathcal{D} \cup \{(s_t, a_t, r(s_t, a_t), s_{t+1})\}$
7:    **end for**
8:    **for** each gradient step **do**
9:       $\theta \leftarrow \theta - \lambda_q \nabla_\theta \hat{J}_q(\theta)$
10:      $\phi \leftarrow \phi + \lambda_\pi \nabla_\phi \hat{J}_\pi(\phi)$.
11:      $\theta^* \leftarrow \tau\theta + (1-\tau)\theta^*$
12:    **end for**
13: **end for**

---

## L EXPERIMENTS RESULTS

### L.1 UNCERTAINTY-AWARE REGULARIZATION EFFECT VIA ABLATION STUDY IN ACTOR CRITIC

We study the uncertainty-aware regularization effect from being categorical distributional in the actor-critic framework, where we decompose the C51 critic loss in distributional SAC (DSAC) according to Eq. 2. We denote the decomposed DSAC (C51) with different $\varepsilon$ as $\mathcal{H}(\mu, q_\theta)(\varepsilon = 0.8/0.5/0.1)$. As suggested in Figure 4, the performance of $\mathcal{H}(\mu, q_\theta)$ tends to vary from the vanilla DSAC (C51) to SAC with the decreasing of $\varepsilon$ on three MuJoCo environments, except bipedalwalk-

Table 1: Hyper-parameters Sheet.

| Hyperparameter | Value |
|---|---|
| *Shared* | |
|    Policy network learning rate | 3e-4 |
|    (Quantile) Value network learning rate | 3e-4 |
|    Optimization | Adam |
|    Discount factor | 0.99 |
|    Target smoothing | 5e-3 |
|    Batch size | 256 |
|    Replay buffer size | 1e6 |
|    Minimum steps before training | 1e4 |
| *DSAC with C51* | |
|    Number of Atoms ($N$) | 51 |
| *DSAC with IQN* | |
|    Number of quantile fractions ($N$) | 32 |
|    Quantile fraction embedding size | 64 |
|    Huber regression threshold | 1 |

| Hyperparameter | Temperature Parameter $\beta$ | Max episode lenght |
|---|---|---|
| Walker2d-v2 | 0.2 | 1000 |
| Swimmer-v2 | 0.2 | 1000 |
| Reacher-v2 | 0.2 | 1000 |
| Ant-v2 | 0.2 | 1000 |
| HalfCheetah-v2 | 0.2 | 1000 |
| Humanoid-v2 | 0.05 | 1000 |
| HumanoidStandup-v2 | 0.05 | 1000 |
| BipedalWalkerHardcore-v2 | 0.002 | 2000 |

erhardcore. In bipedalwalkerhardcore. this tendency may not be clear, as we hypothesis that the algorithm performance is not sensitive when $\varepsilon$ changes within this restricted range, although this range is designed to guarantee a valid density decomposition. It is worth noting that our return density decomposition is valid only when $\epsilon \geq 1 - p_E$ as shown in Proposition 1, and therefore $\epsilon$ can not strictly go to 0, where $\mathcal{H}(\mu, q_\theta)$ would degenerate to SAC ideally. In addition, compared with the ablation study in Figure 3, the trend varying from DSAC to SAC by decreasing $\varepsilon$ may not be as pronounced as that in value-based RL evaluated on Atari games. This is because the actor-critic architecture is generally perceived to be more prone to instability compared to value-based learning in RL. As outlined in (Fujimoto et al., 2018), this instability stems from the policy updates, which may introduce additional bias or variance from the critic learning process.

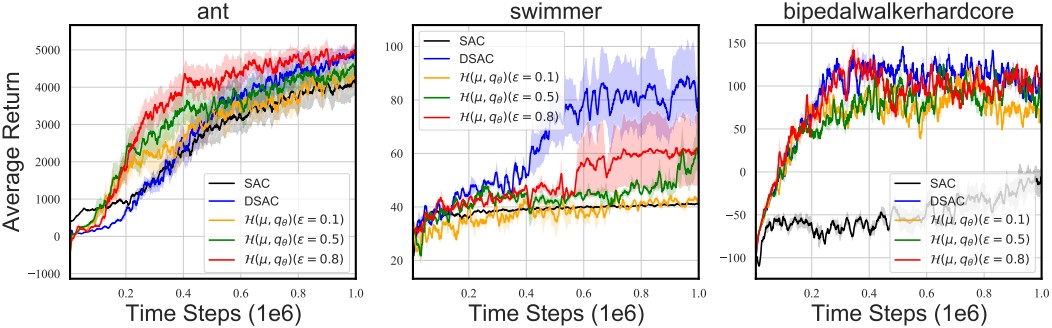

Figure 7: Learning curves of Distributional AC (C51) with the return distribution decomposition $\mathcal{H}(\mu, q_\theta)$ under different $\varepsilon$.

### L.2 SENSITIVITY ANALYSIS OF DERAC

Figure 8 shows that DERAC with different $\lambda$ in Eq. 9 may behave differently in different environments. In general, DERAC with different $\varepsilon$ and $\lambda$ perform similarly to DERAC, with an interpolation nature between AC and DAC (C51). Notably, DERAC with different $\varepsilon$ and $\lambda$ still surpasses at both AC and DAC (C51) in bidedalwalkerhardcore, demonstrating the robust superiority of DERAC algorithm.

### L.3 MUTUAL IMPACTS ON DSAC (C51)

We presents results on seven MuJoCo environments and omits Bipedalwalkerhardcore due to some engineering issue when the C51 algorithm interacts with the simulator. Figures 9 showcases that the simultaneous leverage of uncertainty-aware and vanilla entropy regularization renders a mutual improvement on humanoidstandup and Walker2d. In contrast, the two regularization when em-

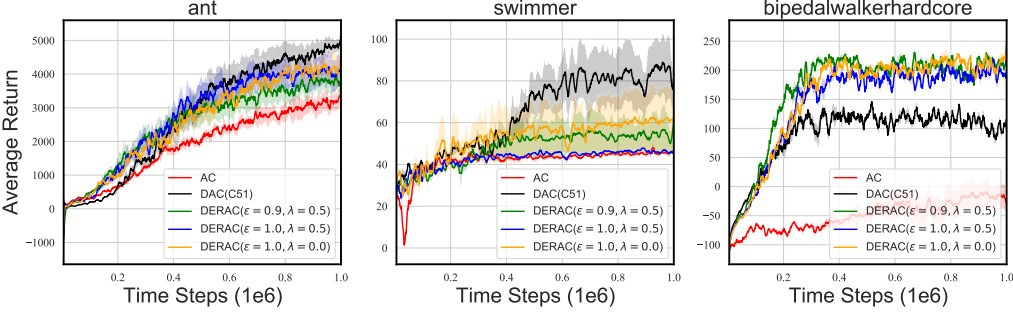

Figure 8: Learning curves of DERAC algorithms across different $\lambda$ and $\varepsilon$ on three MuJoCo environments over 5 seeds.

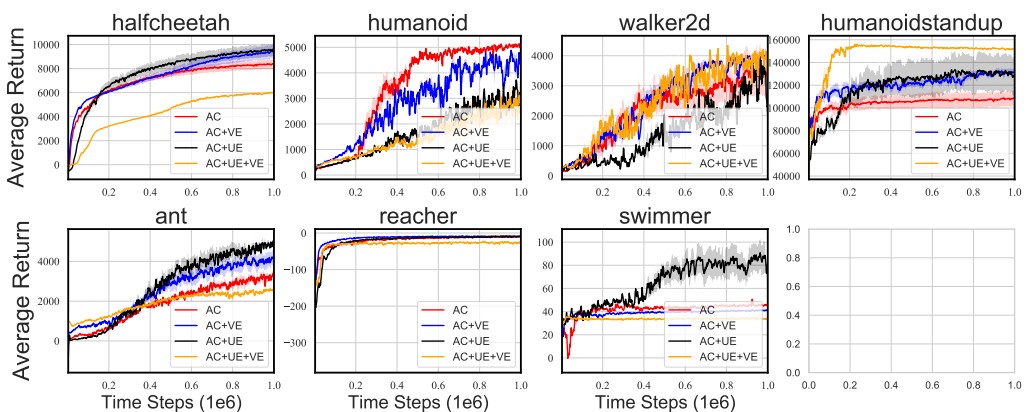

Figure 9: Learning curves of *AC*, *AC+VE* (SAC), ***AC+UE*** (DAC) and AC+UE+VE (DSAC) over 5 seeds across seven MuJoCo environments where distributional RL part is based on C51. (**Walker 2d and Humanoidstandup**): Mutual Improvement. (**Others**): Potential Interference.

ployed together lead to a performance degradation in other environments, especially in swimmer and halfcheetah, where *AC+UE+VE* is significantly inferior to ***AC+UE*** or *AC+VE*.

### L.4 ABLATION STUDY ACROSS DIFFERENT BIN SIZES (NUMBER OF ATOMS)

To further demonstrate our regularization effect based on the return density decomposition, we conducted an additional ablation study by varying the number of bins / atoms (equivalent to adjusting the bin sizes) of both C51 and our decompose algorithm $\mathcal{H}(\mu, q_\theta)$. Consistent with the tendency shown in Figure 3 in Section 6.1, Figure 10 also suggests that decreasing $\varepsilon$ implies that $\mathcal{H}(\mu, q_\theta)$ degrades from C51 with the same bin size to DQN. Another interesting observation is that, as shown in Breakout (the first row in Figure 10), increasing the number of atoms (reducing the bin size) restricts the range of $\epsilon$ for a valid return density decomposition in Proposition 1. Consequently, a small number of atoms or a large bin size can allow a broader variation of $\mathcal{H}(\mu, q_\theta)$ from C51 to DQN, facilitating the demonstration of our regularization effect empirically.

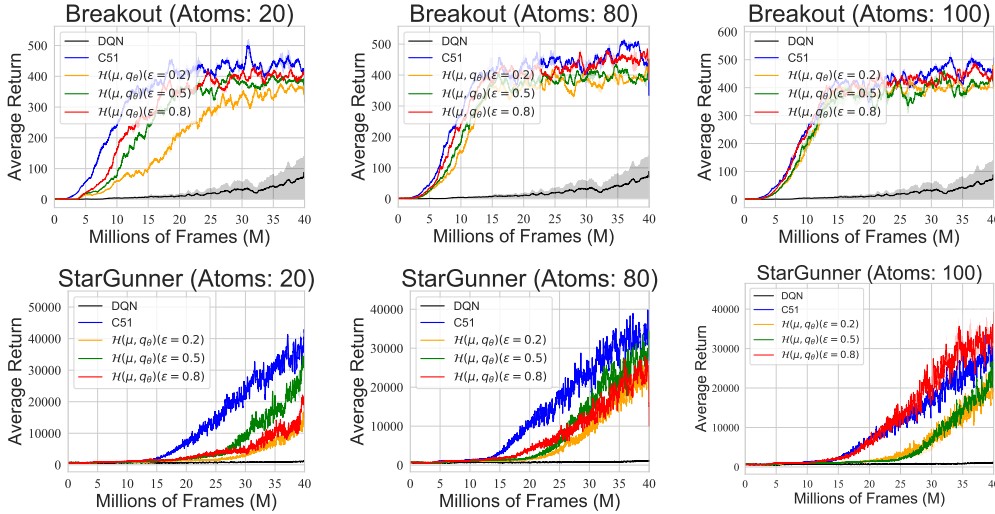

Figure 10: Learning curves of value-based CDRL, i.e., C51 algorithm, and the decomposed algorithm $\mathcal{H}(\mu, q_\theta)$ **across different numbers of atoms (various bin sizes)** on two Atari games. Results are averaged over 3 seeds and the shade represents the standard deviation.

# M    DISCUSSION ON DECOMPOSING QUANTILE-BASED DISTRIBUTIONAL RL

In this section, we discuss about how to decompose the quantile-based distributional loss in quantile regression distributional RL. In each phase of Neural FZI, we know that the return distribution, typically also parameterized by quantiles, is fixed. This, therefore, leads to a composite quantile loss (Zou & Yuan, 2008):

$$\ell_{\text{quantile}} = \frac{1}{N} \sum_{i=1}^{N} \mathbb{E}_{y \sim P_Y} \left[ \rho_{\tau_i} \left( y - Z_\theta^{\tau_i} \right) \right], \tag{42}$$

where we use $P_Y$ to denote the fixed target return distribution. *In quantile-based distributional RL, we can directly sample $y$ from the quantile function $F_Y^{-1}$ of the fixed target return as both the current and target return distributions are parameterized by the quantiles.* $Z_\theta^{\tau_i}$ represents the $\tau_i$-quantile value of the current return distribution. $\rho_{\tau_i}$ can be the vanilla quantile loss defined by: $\rho_{\tau_i}(u) = u \left( \tau_i - \delta_{\{u<0\}} \right), \forall u \in \mathbb{R}$. Alternatively, $\rho_{\tau_i}$ can be the quantiel Huber loss (Huber, 1992), a smooth version of vanilla quantile loss at zero, by additionally introducing a hyper-parameter $\kappa$. We thus denote the quantile Huber loss as $\rho_{\tau_i}^\kappa$, which is defined as:

$$\rho_{\tau_i}^\kappa(u) = \left| \tau_i - \mathbb{1}_{\{u<0\}} \right| \frac{\mathcal{L}_\kappa(u)}{\kappa}, \tag{43}$$

where

$$\mathcal{L}_\kappa(u) = \begin{cases} \frac{1}{2} u^2, & \text{if } |u| \leq \kappa \\ \kappa \left( |u| - \frac{1}{2}\kappa \right), & \text{otherwise} \end{cases}. \tag{44}$$

As $\kappa \to 0$, quantile Huber loss reverts to the vanilla quantile loss. To simplify the notation, we consider the inner-level loss for a fixed $y$:

$$L_{\text{quantile}} = \frac{1}{N} \sum_{i=1}^{N} \rho_{\tau_i} \left( y - Z_\theta^{\tau_i} \right). \tag{45}$$

Unlike normal representation and categorical represent with a proper projection to satisfy the mean-preserving property, quantile distributional dynamic programming is generally not mean-preserving, as the quantiles are non-linear functionals of distribution (Bellemare et al., 2023). However, we show that the quantile representation has an asymptotic connection with the mean-preserving property as *the mean of quantiles is asymptotically equivalent to the expectation of the considered distribution when the number of quantiles tends to infinity.* Assume that we have $N$ evenly spaced quantiles, we approximate the expectation by the mean of the all quantiles values defined by

$$\frac{1}{N} \sum_{i=1}^{N} F^{-1}(\frac{i}{N+1}). \tag{46}$$

Consequently, given a random variable $X$ with its quantile function $F^{-1}$, we have the following property of quantile function:

$$\lim_{N \to +\infty} \frac{1}{N} \sum_{i=1}^{N} F^{-1}(\frac{i}{N+1}) = \int_0^1 F^{-1}(\tau) d\tau = \int_{-\infty}^{+\infty} x \, dF(x) = \mathbb{E}[X], \tag{47}$$

where the first equation results from the relationship between the limit of Riemann Sum and its integral, and the second equation holds by changing the variable $\tau = F(x)$. Note that this asymptotic regime is similar to that in our histogram function analysis for CDRL, where $\Delta \to 0 \iff N \to +\infty$. According to this equivalence regarding the mean quantiles and the expectation, we consider the following two decomposition ways.

**Decomposition Method 1.** We denote $\bar{Z} = \frac{1}{N} \sum_{i=1}^{N} Z_\theta^{\tau_i}$ as the mean of the quantiles for the *current* return. Consequently, we have a straightforward composition as follows:

$$\rho_{\tau_i} \left( (y - \bar{Z}) + \left( \bar{Z} - Z_\theta^{\tau_i} \right) \right) = \rho_{\tau_i}(y - \bar{Z}) + \delta_\tau, \tag{48}$$

where

$$\delta_\tau = \rho_{\tau_i} \left( (y - \bar{Z}) + \left( \bar{Z} - Z_\theta^{\tau_i} \right) \right) - \rho_{\tau_i}(y - \bar{Z}). \tag{49}$$

Therefore, we have the *decomposed* composite quantile loss as

$$L_{\text{quantile}} = \underbrace{\frac{1}{N}\sum_{i=1}^{N}\rho_{\tau_i}(y - \bar{Z})}_{\text{Mean-Related Term}} + \underbrace{\frac{1}{N}\sum_{i=1}^{N}\delta_\tau}_{\text{Residual Term}}. \tag{50}$$

The first term is a mean-related term, which we will elaborate later, while the induced $\delta_\tau$ in the residual term is aimed at capturing the distribution information beyond only the expectation. Particularly, minimizing $\rho_{\tau_i}\left((y - \bar{Z}) + (\bar{Z} - Z_\theta^{\tau_i})\right)$ in $\delta_\tau$ will push the deviations $\bar{Z} - Z_\theta^{\tau_i}$ from the current return estimator to capture the deviations from the target return distribution $y - \bar{Z}$. This regularization term contributes to preserving the richness of the quantile representation for distributional information from the return.

In terms of the mean-related term, let us consider the approximation. As the quantile Huber loss is typically used in quantile-based distributional RL, when $\kappa$ is large, the mean-related term can be simplified as

$$\frac{1}{N}\sum_{i=1}^{N}\rho_{\tau_i}(y - \bar{Z}) = \frac{1}{N}\sum_{i=1}^{N}\left|\tau_i - \mathbb{1}_{\{y - \bar{Z} < 0\}}\right|\frac{1}{2}(y - \bar{Z})^2 \approx \frac{1}{4}(y - \bar{Z})^2, \tag{51}$$

where the approximation holds because $\left|\tau_i - \mathbb{1}_{\{y - \bar{Z} < 0\}}\right|\frac{1}{2}(y - \bar{Z})^2$ is just the quantile value scaled version of least squared loss. Since $\bar{Z}$ is the expectation of all quantiles, it can be approximately symmetric to $\mathbb{E}[Y]$. Suppose $P(y - \bar{Z} < 0) = P(y - \bar{Z} > 0) = \frac{1}{2}$, we have

$$\mathbb{E}\left[\left|\tau_i - \mathbb{1}_{\{y - \bar{Z} < 0\}}\right|\right] = \frac{1}{2}\left(|\tau_i - 1| + \tau_i\right) = \frac{1}{2}. \tag{52}$$

Therefore, this approximation in the mean-related term holds. This implies that the mean quantile estimator $\widehat{\bar{Z}}$ captures the expectation of the target return distribution from $y \sim P_Y$. Recap the asymptotic equivalence between the expected quantiles and the expectation, the limiting estimator of $\widehat{\bar{Z}}$ *by minimizing the mean-related term in $\ell_{quantile}$* satisfies the following equation:

$$\widehat{\bar{Z}} = \mathbb{E}[Y], \quad \text{and} \quad \lim_{N \to \infty}\widehat{\bar{Z}} = \mathbb{E}[Z_\theta], \tag{53}$$

where $\mathbb{E}[Z_\theta]$ is the expectation of the current return $Z_\theta$.

In summary, the first decomposition method decomposes the quantile-base distributional loss into the mean-related and residual terms. After a mild approximation, the mean-related term can be simplified as a least-squared loss equipped with an expected quantiles estimator. Combining the equivalence regarding the limiting behavior of the expected quantiles, *the mean-related term is thus approximately equivalent to the standard least-squared loss used in classical RL, asymptotically satisfying the mean-preserving property in distributional dynamic programming*. Moreover, the residual term is able to capture the return distribution information beyond its expectation. In the context of uncertain-aware regularized exploration in our paper, the residual term plays the similar role of the cross-entropy-based regularization derived in Proposition 2 of CDRL.

**Decomposition Method 2.** The other decomposition method can directly follow the return density decomposition proposed in Eq. 2, but we apply the decomposition on the quantile function $F^{-1}(\tau)$ for $\tau \in [0, 1]$. We expect that this decomposition also leads to two parts, where the first part can involve the quantile defined on the a bin $\Delta_{\bar{\tau}}$ that contains the expected quantiles $\bar{F}^{-1} = \sum_{i=1}^{N}F^{-1}(\tau_i)$, and the second term relates to the distribution part. However, this decomposition is largely beyond the existing techniques we proposed in this paper, and it takes more efforts to think it carefully. We leave this decomposition regarding the quantile function as future work.

