# OpenReview forum: "The Benefits of Being Categorical Distributional: Uncertainty-aware Regularized Exploration in Reinforcement Learning"
_ICLR.cc/2025/Conference — Submitted to ICLR 2025_

### Official Review · Reviewer_hCtQ · 2024-10-31

**Soundness:** 3
**Presentation:** 2
**Contribution:** 2
**Rating:** 3
**Confidence:** 3

**Summary:**

This paper examines the success of categorical distributional RL algorithms by formulating the distributional component of these algorithms as a form of regularization. The authors support this perspective through theoretical analysis. It is also followed by experiments to validate their findings.

**Strengths:**

- This paper provides theoretical explanations on distributional RL (concretely, on categorial distributional RL algorithms). It is an important problem.

- This paper explores both the theoretical aspects (considering the regularization effect) and empirical validation, making it relatively comprehensive overall.

**Weaknesses:**

- The paper is not easy to follow. Many words are vague. This is especially noticeable in section 4.2. For instance, the phrase "characterizing the impact of action-state return distribution beyond its expectation"  is vague to me and may need explanation (Line 196)
- There have been prior works on the theory of distributional RL, including those referenced around Line 50 (Rowland et al, 2018, 2023, 2024). Therefore, the statement in the abstract is not very accurate: "Despite the remarkable empirical performance of distributional reinforcement learning (RL), its theoretical advantages over classical RL remain elusive."
- When reading Line 55, I found the authors mentioning "Yet, their findings are not directly based on typical distributional RL algorithms commonly used in practice, such as C51 or QR-DQN". This implies that this authors is probably going to analyze a typical distribution RL algo like C51. However,  the paper instead focuses on DERPI and develops DERAC in section 5.2. Both are different from C51 (although they have some similarities). Thus, I feel (1) Line 55 is not precise enough and (2) the paper's scope is somewhat narrow, given that categorical distributional RL is already a specific subset within the field.
- I have more concerns regarding novelty that I listed in the questions section.

**Questions:**

The theoretical contributions appear to have limited novelty in light of prior work. For instance, proposition 3 looks pretty close to prior results on the convergence of categorial algorithm (such as in [1]) ? We already know such convergence results when $\Delta$ goes to zero. Could you clarify the novelty here?

[1] Bellemare, Marc G., Will Dabney, and Mark Rowland. *Distributional reinforcement learning*. MIT Press, 2023.

---

> ### Author Response · Authors · 2024-11-20
>
> We would like to express our gratitude for your time and effort in reviewing our paper. We really appreciate the insightful and detailed comments and suggestions you have provided, and we would like to address each comment in the section on Weaknesses and Questions you raised in your review.
>
> ### Weakness
>
> >1. The paper is not easy to follow. Many words are vague. This is especially noticeable in section 4.2. For instance, the phrase "characterizing the impact of action-state return distribution beyond its expectation" is vague to me and may need explanation (Line 196)
>
> Thank you for highlighting this clarity issue. Based on your suggestion, we have revised this statement in the updated paper to better explain the impact of the induced histogram function on the behavior difference between distributional and classical RL. We have also thoroughly proofread the entire paper to improve clarity, and we would appreciate any further feedback if there are still vague statements or terms.
>
> >2. There have been prior works on the theory of distributional RL, including those referenced around Line 50 (Rowland et al, 2018, 2023, 2024). Therefore, the statement in the abstract is not very accurate: "Despite the remarkable empirical performance of distributional reinforcement learning (RL), its theoretical advantages over classical RL remain elusive."
>
> We apologize for the statement in the abstract that may have been inaccurate. Based on your suggestion, we have revised it to 'Despite the remarkable empirical performance of distributional reinforcement learning (RL), its theoretical advantages over classical RL **are still not fully understood**.'
>
> >3. When reading Line 55, I found the authors mentioning "Yet, their findings are not directly based on typical distributional RL algorithms commonly used in practice, such as C51 or QR-DQN". This implies that this authors is probably going to analyze a typical distribution RL algo like C51. However, the paper instead focuses on DERPI and develops DERAC in section 5.2. Both are different from C51 (although they have some similarities). Thus, I feel (1) Line 55 is not precise enough and (2) the paper's scope is somewhat narrow, given that categorical distributional RL is already a specific subset within the field.
>
> We want to clarify that we indeed study C51 and investigate its behavior differences from DQN using return density decomposition, particularly in Section 4. Section 5 further explores the decomposed regularization effect within the policy gradient framework, and, as a result, we introduce DERPI and DERAC **as byproduct algorithms to guarantee the convergence when incorporating the uncertain-aware regularization term**. As also mentioned in Section 6.2, the primary purpose of introducing DERAC is to interpret the benefits of CDRL from the perspective of uncertain-aware regularized exploration rather than to pursue empirical superiority. Therefore, while the two follow-up algorithms are proposed following our theoretical analysis, **our main goal remains to interpret the benefits of the practical CDRL algorithm**.
>
> Although CDRL is a specific class of distributional RL, it is recognized as the first successful implementation in distributional RL, with broad recognition in the community. Unfortunately, the benefits of CDRL are not yet fully understood by the current research community. Therefore, it is valuable to focus on CDRL for a rigorous analysis as the first step before extending our analysis to other algorithmic classes. To further enhance our contribution, we have provided **a discussion about how to decompose quantile-based distributional RL in Appendix M** of the revised paper. We invite the reviewer to refer to this discussion, and we welcome any further feedback.

---

> > ### Author Response · Authors · 2024-11-20
> >
> > ### Questions
> >
> > >The theoretical contributions appear to have limited novelty in light of prior work. For instance, proposition 3 looks pretty close to prior results on the convergence of categorial algorithm (such as in [1]) ? We already know such convergence results when $\Delta$ goes to zero. Could you clarify the novelty here?
> >
> > Existing convergence results for categorical distributional RL, such as [1] and Theorem 4.3.2 in [2], primarily focus on the Bellman update by using the **complete** categorical distribution. By contrast, Proposition 3 examines the minimizer of the  **mean-related term (a) decomposed from**  categorical distribution within a supervised loss framework instead of concerning the full distribution, thereby establishing a connection between the objective functions of CDRL and classical RL. Based on this equivalence, we can then attribute the distribution-matching regularization as the primary source of behavior differences between classical and distributional RL. Consequently, Proposition 3 and existing convergence results in CDRL are **from distinct perspectives and serve complimentary purposes, highlighting fundamental differences**. Importantly, Proposition 3 results from the return density decomposition, a novel technique not previously proposed. That being said, the asymptotic equivalence between objective functions of CDRL and classical RL, as established in Proposition 3, is indeed a novel contribution.
> >
> >
> > In summary, we express our gratitude to the reviewer for their consistent dedication to reviewing our work. Should this rebuttal addresses your concerns, we would be grateful for a revised score. Of course, we remain at your disposal for any further clarifications.
> >
> >
> > ### Reference
> >
> > [1] Bellemare, Marc G., Will Dabney, and Mark Rowland. Distributional reinforcement learning. MIT Press, 2023.
> >
> > [2] Rowland, Mark, et al. "An analysis of categorical distributional reinforcement learning." AISTATS, 2018.

---

> > > ### Author Response · Authors · 2024-11-23
> > >
> > > Thank you for your dedication and interest in our paper. As the author and reviewer discussion period approaches its end, we are curious to know your thoughts on our rebuttal and whether you have any additional questions. Should this rebuttal address your concerns, we would be grateful for a revised score. Of course, we remain at your disposal for any further clarifications.

---

> > > > ### Comment · Reviewer_hCtQ · 2024-11-25
> > > >
> > > > I appreciate the authors’ detailed response. However, I still think the paper could be clearer, and the contribution should be clarified more. Moreover, the authors describe DERPI and DERAC as “byproduct algorithms” in the response but still, the paper centers on them, which appear to be different from classic categorial algorithms and thus kinda narrow.

---

> > > > > ### Author Response · Authors · 2024-11-25
> > > > >
> > > > > Thanks for your timely response. We would like to clarify further that the primary goal of this paper is ALWAYS to interpret the benefits of using categorical distribution in RL. To achieve this goal, we first explore this benefit in value-based RL in Section 4 and then we extend our analysis to policy gradient-based RL in Section 5, particularly Section 5.1. The DERAC and DERPI algorithms are introduced in **a separate section in Section 5.2 with only half of one page**, which composes **a separate algorithmic contribution**.
> > > > >
> > > > > ## Contribution and Outline
> > > > >
> > > > > In fact, we apologize if you did not fully recognize our paper's structure, but we have already provided the outline at the end of Section 1 in the initially submitted paper, which we believe has made the logic of our paper clear by our best efforts.
> > > > >
> > > > > ## Role of DERAC and DERPI
> > > > >
> > > > > Note that the explanation regarding the role of DERAC and DERPI is not just given in the response but **has already been provided in the original submission**, e.g., in the second bullet point at the end of the Introduction part (Lines 80 - 85), where we state that 'A byproduct interpretable algorithm is further introduced, interpolating between classical and distributional RL.' This clarification has also been reiterated in the experimental part (Section 6.2) **still in the originally submitted paper**, saying that 'the primary purpose of introducing DERAC is to interpret the benefits of CDRL from the perspective of uncertain-aware regularized exploration, rather than to pursue the empirical superiority.'
> > > > >
> > > > > In summary, most parts of our paper (except Section 5.2, which provides an independent algorithmic contribution for the comprehensive purpose) are **EXACTLY studying the benefits of classical categorical algorithms (CDRL)**. In addition, we have also provided a follow-up discussion about how to extend our analysis in CDRL to **quantile-based distributional RL in Appendix M, further broadening the scope and contribution of our work**.
> > > > >
> > > > > We sincerely hope the reviewer can re-evaluate the actual contribution of our paper. Of course, we are happy to provide any additional clarifications if needed.

---

### Official Review · Reviewer_U83M · 2024-11-02

**Soundness:** 3
**Presentation:** 3
**Contribution:** 3
**Rating:** 8
**Confidence:** 2

**Summary:**

This paper provides an explanation of how the Categorical Distributional RL (CDRL) outperforms classical RL. Essentially, CDRL contains an implicit entropy regularization that encourages exploration through visiting states where the environment uncertainty is largely underestimated. This uncertainty-aware regularization provides context about the superiority of distributional RL.

**Strengths:**

Authors neatly derived the uncertainty-aware regularization term in distributional RL that is responsible for their superiority. Furthermore, an explicit demonstration of the link between distributional and classic RL through regulating the effects of the uncertainty-aware regularization term is provided.

**Weaknesses:**

1) Histogram function estimator is sensitive to bin widths, yet it is unclear how the bin widths in the experiments were selected. I suggest that an ablation study varying the bin widths be carried and report how it impacts the results - this would be informative.

2) Please provide intermediate steps in equation (29) for proof of proposition 2 that shows how the constant $\Delta^{-2}$ disappeared. It currently unclear and would be great if it is explicitly shown.

**Questions:**

1) How can the $p_{E}$ (coefficient on bin $\Delta_{E}$) be computed?
2) What are the effects of various bin sizes on the results?

---

> ### Author Response · Authors · 2024-11-20
>
> Thank you for taking the time to review our paper. We appreciate your positive assessment and insightful feedback, and we would like to address the concerns you raised in your review.
>
> ### Weakness
>
> >1. Histogram function estimator is sensitive to bin widths, yet it is unclear how the bin widths in the experiments were selected. I suggest that an ablation study varying the bin widths be carried and report how it impacts the results - this would be informative.
>
> The C51[1] is a typical CDRL algorithm, which uses 51 atoms (equivalent to 51 bins). We thus also use 51 bins in our algorithm implementation to maintain the original favorable performance of C51. To further address your concern, **we have conducted an ablation study by varying the bin sizes (number of atoms) in two Atari games, with the results presented in Figure 10 of Appendix L.4**. These results suggest that the performance of the new ablation study is consistent with our reported results in Section 6.1. Therefore, our conclusion regarding the regularization effect from the return density decomposition is robust against different bin sizes.
>
>
> >2.  Please provide intermediate steps in equation (29) for proof of proposition 2 that shows how the constant disappeared. It currently unclear and would be great if it is explicitly shown.
>
> We would like to clarify that in Eq. 29 (Eq 28 in the updated version), $1(x\in\Delta^i_E) / \Delta   \log q_\theta^{s_i, a_i}(\Delta_j) / \Delta$ represents $\frac{1(x\in\Delta^i_E)}{\Delta}  \log \frac{q_\theta^{s_i, a_i}(\Delta_j)}{\Delta}$ and we apologize for this potential confusion. To eliminate this ambiguity, we updated this formula in the revised paper. Regarding how $\Delta^{-2}$ disappears, the first $\frac{1}{\Delta}$ disappears as we take the integral $\int_{l_i}^{l_{i+1}}$. The second one is within the $\log$, and serves as a constant in the end, which does not affect the optimization.
>
> ### Questions
>
> >1. How can the p_E (coefficient on bin \Delta_E) be computed?
>
> Given the target return distribution of $Y_i^k$, which is fixed in each phase of Neural FZI, we can evaluate the expectation of the histogram density function by $E[Y_i^k] = \sum_{i=0}^{N-1} \frac{l_l + l_{i+1}}{2} p_i.$ Then we index the bin $\Delta_E$ that contains $E[Y_i^k]$. Consequently, the probability (the height times the bin size) on that bin is $p_E$. This computation process is also consistent with what we have done in our implementation.
>
> >2. What are the effects of various bin sizes on the results?
>
> Please refer to the response to Weakness 1.
>
>
> In summary, thank you again for pointing to these potential areas of improvement. We very much appreciate your suggestions. Please let us know if you have any further comment or feedback.
>
> ### Reference
>
> [1] Bellemare, Marc G., Will Dabney, and Rémi Munos. "A distributional perspective on reinforcement learning." International conference on machine learning. PMLR, 2017.

---

> > ### Comment · Reviewer_U83M · 2024-11-21
> >
> > Thanks for taking the time to address the concerns I raised. I like the updates and will improve my rating.

---

> > > ### Author Response · Authors · 2024-11-21
> > >
> > > We are delighted that you found the updates helpful. Thank you very much for your constructive feedback and improved rating.

---

### Official Review · Reviewer_bVVf · 2024-11-03

**Soundness:** 3
**Presentation:** 3
**Contribution:** 2
**Rating:** 6
**Confidence:** 3

**Summary:**

This paper presents a novel decomposition of CDRL’s objective function into an expectation-based term and a distribution-matching regularization. The authors argue that the latter term provide an augmented reward signal, contributing to implicit regularized exploration. Unlike the explicit entropy regularization in MaxEnt RL, this approach encourages exploration aligned with environmental uncertainty. The paper further introduces the Distribution-Entropy-Regularized Actor Critic (DERAC) algorithm, which balances between expectation-based and distributional learning, supported by experiments in Atari and MuJoCo environments.

**Strengths:**

This paper provides a new perspective on interpreting distributional RL by leveraging the decomposition of the action-value distributions. The derivation of uncertainty-aware regularization from the decomposition of CDRL’s return distribution is an innovative contribution that highlights the advantages of distributional RL over traditional methods.

The comparison between the proposed uncertainty-aware regularization and entropy regularization in MaxEnt RL offers valuable insights into the added benefits of distributional RL.

The experiments are solid and provide strong empirical evidence supporting the paper’s theoretical claims.

**Weaknesses:**

The analysis in this paper focuses solely on categorical distributional reinforcement learning (C51). Given that other distributional learning methods, such as QR-DQN and IQN, are becoming more popular, this appears to be a weak point.  It would be helpful for the authors to discuss how their analysis might extend to or differ from other popular distributional RL methods like QR-DQN and IQN.

**Questions:**

Some theoretical results are achieved when $\Delta$ (the bin size) tends to zero. As $\Delta$ approaches zero, $P_E$ typically approaches zero. What happens to the hyperparameter $\epsilon$ in Proposition 1 and $\alpha$ in Equation (4) of Propositions 2 and 3 in the limit case?  It is stated on page 4 that “Proposition 1 demonstrates that the return density distribution is valid when the hyperparameter $\epsilon$ is specified as $\epsilon \ge 1-P_E$.”  Does this imply that $\epsilon$ needs to be close to $1$ when $P_E$ is very small?

In Section 7, the authors mention the potential to extend their work to other distributional RL methods, such as QR-DQN. Given that distributional dynamic programming with quantile representation is not mean-preserving, the extension to QR-DQN may be problematic. The authors need to address why they believe their approach could extend to QR-DQN and whether they have considered using a normal representation instead. Distributional dynamic programming with a normal representation is mean-preserving, making it a more suitable candidate for extending this work.

Minor comment: On page 7, the first sentence after Lemma 1 likely should use "retain" instead of "remain”.

---

> ### Author Response · Authors · 2024-11-20
>
> Thank you very much for taking the time to review our paper. We appreciate your positive assessment and insightful feedback, and we would like to address the concerns you raised in your review.
>
>
> >Question 1: Some theoretical results are achieved when $\Delta$ (the bin size) tends to zero. As $\Delta$ approaches zero, $P_E$ typically approaches zero. What happens to the hyperparameter $\epsilon$ in Proposition 1 and $\alpha$ in Equation (4) of Propositions 2 and 3 in the limit case? It is stated on page 4 that “Proposition 1 demonstrates that the return density distribution is valid when the hyperparameter $\epsilon$ is specified as $\epsilon \geq 1 - P_E$. Does this imply that $\epsilon$ needs to be close to 1 when $P_E$ is very small?
>
> We acknowledge that as $\Delta \rightarrow 0$, $P_E \rightarrow 0$ and $\epsilon \rightarrow 1$ ($\alpha \rightarrow \infty$) to ensure a valid histogram function, satisfying $\epsilon \geq 1 - p_E$ in the return density decomposition. Note that this proposed return density decomposition is valid for any non-trivial $\epsilon$ as long as $\epsilon \geq 1 - p_E$, which allows for developing practical algorithms if needed. In contrast, the results presented in Proposition 2 and 3 provide an **asymptotically approximated** connection between optimizing the term (a) in the decomposed distribution loss in Eq.3 (the updated paper) and optimizing the classical RL objective function.
>
> We understand the review may be concerned about the gap between the asymptotic connection in our analysis with the practical algorithm implementation. In fact, we have considered this potential concern in the submitted paper and have provided **the rate of convergence $o(\Delta)$** in terms of the absolute difference of cumulative distribution function (CDF) between the converging CDF of term (a) and the step function (the CDF of a Dirac Delta function) in Appendix G. In the process of $\Delta \rightarrow 0$, the minimizer of optimizing term (a) gradually tends to that of optimizing classical RL, **with the error shrinking at the rate of $o(\Delta)$**. Please review Appendix G for more details.
>
>
> >Question 2: In Section 7, the authors mention the potential to extend their work to other distributional RL methods, such as QR-DQN. Given that distributional dynamic programming with quantile representation is not mean-preserving, the extension to QR-DQN may be problematic. The authors need to address why they believe their approach could extend to QR-DQN and whether they have considered using a normal representation instead. Distributional dynamic programming with a normal representation is mean-preserving, making it a more suitable candidate for extending this work.
>
> Thank you for raising this interesting and technical point regarding our future work. We still believe that our analysis can be extended quantile-based distributional RL at least partially, though the techniques involved may be different. We acknowledge that distributional dynamic programming with the quantile representation is not $exactly$ mean-preserving as quantiles are non-linear functionals of distribution and focus on capturing the shape of distributions. However, we emphasize the fact that quantile dynamic programming can be **asymptotically mean-preserving** owing to the property that the expected quantiles can converge to the true expectation of the distribution **as the number of quantiles approaches infinity**. For a given random variable $X$, we have
> $$	\lim_{N\rightarrow+\infty} \frac{1}{N} \sum_{i=1}^{N} F^{-1}(\frac{i}{N+1}) = \int_{0}^{1} F^{-1}(\tau) d \tau = \int_{-\infty}^{+\infty} x d F(x) = \mathbb{E}\left[X\right].$$
> Additionally, we provide a more detailed explanation, including **two potential decomposition methods on quantile distributional loss in Appendix M of the revised paper**. We recommend the reviewer to refer to this discussion, where we extend our decomposition method to quantile-based distributional RL. The revised paper thus has enhanced contribution and mitigated the limitation present in the original submission.
>
> Thank you also for bringing the normal representation to our attention. We fully agree with you that using normal representation (Section 5.4 in [1]) can be (theoretically) convenient. However, there still remains a potential gap between the well-behaved normal representation and more practical fixed-size representation (Section 5.5 in [1]), e.g., quantile, categorical distribution, and particles.
>
> We have fixed the typo mentioned in the minor comment in the revised paper.
>
> Overall, thank you again for pointing out these potential areas for improvement. We appreciate your suggestions. Please let us know if you have any further comments or feedback.
>
> ### Reference
>
> [1] Bellemare, Marc G., Will Dabney, and Mark Rowland. Distributional reinforcement learning. MIT Press, 2023.

---

> > ### Author Response · Authors · 2024-11-23
> >
> > Thank you for your dedication and interest in our paper. As the author and reviewer discussion period approaches its end, we are curious to know your thoughts on our rebuttal and whether you have any additional questions. Should this rebuttal address your concerns, we would be grateful for an increased score. Of course, we remain at your disposal for any further clarifications.

---

> > > ### Comment · Reviewer_bVVf · 2024-12-02
> > >
> > > Thank you to the authors for the detailed response, which addresses many of the concerns raised by the reviewers.

---

### Official Review · Reviewer_X6Vg · 2024-11-04

**Soundness:** 1
**Presentation:** 1
**Contribution:** 1
**Rating:** 3
**Confidence:** 5

**Summary:**

This paper studies one of the advantages that distributional reinforcement learning, especially the categorical version, has in executing uncertainty-aware regularized exploration.
The authors claim that categorical distributional reinforcement learning (CDRL) implicitly updates the return distribution and uncertainty simultaneously when we learn the return distribution by value (return distribution) iteration.
The induced uncertainty term can be considered as an entropy regularization which is analogous to maximum entropy RL.
By the proposed analysis, CDRL has a strength in terms of exploration compared to standard RL algorithms.

**Strengths:**

In the research of distributional RL, there are a few analyses of what strengths distributional RL has compared to standard RL.

This paper provides an interesting view that uncertainty of return distribution is additionally learned and provided as the entropy for encouraging exploration.

In addition, this paper provides the experimental results of environments with both continuous and discrete action spaces.

**Weaknesses:**

Despite the above strengths, this paper has the following critical weaknesses.

If I misunderstood and was worng about the following comments, I’ll be happy to discuss with the authors.

1. **Quality of writing**: the main weakness of this paper is the quality of writing, which makes readers hard to understand the main contribution.
    1. Section 3 and 4.1 only covers the superficial backgrounds to understand how return density decomposition can be conducted. For example, there is no detailed explanation on C51, which helps to understand why  $\hat{\mu}$ can be defined.
    2. The notations and definitions are not well presented to make readers understand the main contribution. For example, the definition of $\hat{\mu}$ is followed by Equation (3) without enough statements.
    3. Experiment section is presented without any descripton of baseliens and environemnts.
2. **Not rigorous proof**: the theortical proofs to show the property of categorical distributional RL ar not rigorous to validate the main concept.
    1. There is no explicit proof how the decomposed term in Equation (3) is related to categorical representation. Especially, there is no explanation on the relation between $\mu$ and categorical distribution.
    2. Proposition 2 has too strong approximation that KL divergence is proportional to the cross entropy. This can underestimate the uncertainty of ground-truth distribution.
3. **Weak presentation of experiment section**: Since the author claims that the proposed method is related to exploration, but there is no discussion on exploration schedule and comparison with other exploration schem, such as DLTV. In addition, the proposed method cannot ourperform C51, which only uses the default exploration method, epsilon-greedy.

**Questions:**

What is the main role and exact definition of $\mu$? I can partially agree that the categorical representation can be considered as a histogram density representation, but I cannot understand how $\mu$ can be introduced.

---

> ### Author Response · Authors · 2024-11-20
> **Author Response**
>
> We would like to sincerely express our gratitude for your time and effort spent reviewing our paper. We really appreciate the insightful and detailed comments and suggestions that have been provided. We would like to address each concern you raised in your review and are very happy to discuss them with you further.
>
> ### Weakness: 1. Quality of writing
>
> >1. Section 3 and 4.1 only covers the superficial backgrounds to understand how return density decomposition can be conducted. For example, there is no detailed explanation on C51, which helps to understand why $\mu$ can be defined.
>
> Section 3 covers the basic background and definition of distributional RL, while the Neural FZI in Section 4.1 characterizes the key part of distributional RL, particularly from the perspective of distributional loss. Notably, we **decompose the distributional loss function directly based on Neural FZI in Section 4.2**, when we choose $d_p$ as KL and histogram function to represent $Z_\theta$, which leads to the induced histogram function $\widehat{\mu}$.
>
> We did not introduce too many details of CDRL to save space, as CDRL is the first successful algorithm with broad recognition with the research community. We agree that adding more details of CDRL would be better as these details could cater to more interested readers with more diverse backgrounds beyond distributional RL. To this end, **we have added one paragraph** at the beginning of Section 4.2 before we employ the return distribution decomposition, and **we have added more details, particularly the CDRL update algorithm, in Appendix A of the revised paper**. We believe these revisions can provide readers with sufficient background on CDRL and C51 before we dive deeper into the distributional loss decomposition in Section 4.2.
>
>
> >2. The notations and definitions are not well presented to make readers understand the main contribution. For example, the definition of
>  is followed by Equation (3) without enough statements.
>
> The notation $\widehat{\mu}$ following Eq.3 (Eq. 2 in the revised paper) represents **the induced histogram density function after applying the return density decomposition** for a given histogram density function $\widehat{p}$. To enhance clarity, we have rewritten the notations and definitions in Proposition 2 and Eq.3 in the revised paper, which we believe significantly improves the readability.
>
> We apologize if any other definitions or notations were not well presented. We have carefully proofread our presentation and updated all of them in the revised paper. We sincerely appreciate any further feedback the reviewer can provide regarding specific areas that may still need enhancement.
>
> >3. Experiment section is presented without any descripton of baseliens and environemnts.
>
> We would like to clarify that we employ both Atari games and MuJoCo environments, which are widely used in RL research. In the submitted version, we have stated the environment in both the text and captions of figures in the experimental section.
>
> We have also explained the baselines in the text of each subsection of the experimental section. For example, we have stated how to derive $\mathcal{H}(\mu, q_\theta)$ algorithm in the first paragraph of Section 6.1, and what are the definitions of AC, AC+UE, AC+VE, and AC+UE+VE referring to in the caption of Figure 5.
>
> To further emphasize and differentiate our algorithms, **we provided another subsection in Appendix L.1 of the updated paper, where more detailed implementation and comparison of all algorithms considered are provided**. Although we appreciate this potential clarity concern about baseline algorithms, we believe the updated version has provided a sufficient description of all baselines. Please feel free to let us know if you are still concerned about any considered algorithm in our experiments.

---

> > ### Author Response · Authors · 2024-11-20
> >
> > ### Weakness: 2. Not rigorous proof
> >
> > > 1. There is no explicit proof how the decomposed term in Equation (3) is related to categorical representation. Especially, there is no explanation on the relation between $\mu$ and categorical distribution.
> >
> > Eq.(3) (Eq. 2 in the updated paper) presents an independent decomposition technique we developed to decompose a **histogram density function**. Its connection with categorical distribution is that, as mentioned in the remark immediately following Proposition 1 with proof in Proposition 5 of Appendix C, **the loss function under the histogram function estimator is equivalent to that under the categorical distribution**. This equivalent is intuitive and even straightforward since both methods involve learning the probabilities defined over a fixed set of atoms (support points). We encourage the reviewer to refer to Proposition 5 in Appendix C for a detailed explanation and proof.
> >
> > With this equivalence in mind, the algorithm implementation derived from our analysis is identical to C51, with the only conceptual difference being that we employ a **continuous** histogram density estimator to allow a valid decomposition in the continuous space instead of a **discrete** categorical distribution in C51.
> >
> > Since updating the histogram density function is equivalent to optimizing the categorical distribution in CDRL, $\widehat{\mu}$ remains particularly valid in the continuous space derived from the histogram density function while maintaining a close connection to CDRL in practice.
> >
> > > 2. Proposition 2 has too strong approximation that KL divergence is proportional to the cross entropy. This can underestimate the uncertainty of ground-truth distribution.
> >
> > We would like to clarify that Proposition 2 is exactly derived from the KL-based loss function of Neural FZI in Eq.2, after applying the return density decomposition on the target histogram density function. We acknowledge the potential uncertainty underestimation issue in the general analysis of evaluating KL divergence **if the target density function is not fixed**. However, in our analysis, the KL-based loss is **exactly (not approximately)** proportional to the cross entropy, owing to the use of target network in practical algorithms, which remains fixed during each phase update in Neural FZI.
> >
> > To further address your concern and validate our analysis, we conducted experiments, comparing the performance of C51 with cross-entropy loss against the vanilla C51 equipped with KL divergence, across three Atari games and MuJoCo environments, as shown in **Figure 6 of the new Appendix E.2**. The results indicate that the C51 algorithms with the KL and cross-entropy loss behave similarly across all the environments considered.
> >
> > ### Weakness: 3. Weak presentation of experiment section
> >
> > > Since the author claims that the proposed method is related to exploration, but there is no discussion on exploration schedule and comparison with other exploration schem, such as DLTV. In addition, the proposed method cannot ourperform C51, which only uses the default exploration method, epsilon-greedy.
> >
> > As the title suggests, the main conclusion of our paper is that **we interpret the benefits of using categorical distributional loss as its intrinsic decomposed regularization**, which serves as a new exploration form in RL context. Our objective is not to propose a new exploration method. Instead, we particularly compare our uncertain-aware regularized exploration with MaxEnt RL to help the community better understand the underlying mechanism of using a distributional loss. That is why we believe there is no need to compare with other exploration schedules in RL.
> >
> > DLTV utilizes the variance from the learned return distribution in distributional RL as a bonus term in selecting the action of behavior policy, offering an improved version of existing distributional RL algorithms through the lens of optimistic exploration. By contrast, although both DLTV and our work involve distributional RL and exploration **conceptually**, our focus is on interpreting the benefits of distributional learning as an intrinsic exploration effect from the distributional loss.
> >
> > We would like to clarify that the proposed algorithm in Section 6.1 serves as an ablation study, demonstrating that relying less on the distribution information within distributional RL (C51) causes the algorithm's performance to degenerate to classical RL (DQN). In Section 6.2, as we stated in the experiments, **the primary purpose of introducing DERAC is to interpret the benefits of CDRL from the perspective of uncertain-aware regularized exploration, rather than to pursue the empirical superiority**. Moreover, we observed an interesting result: the interpolated algorithm potentially performs better in complicated environments, such as bipedalwalkerhardcore in Figure 4, mitigating the over-exploration issue.

---

> > > ### Author Response · Authors · 2024-11-20
> > >
> > > ### Questions
> > >
> > > > What is the main role and exact definition of $\mu$? I can partially agree that the categorical representation can be considered as a histogram density representation, but I cannot understand how $\mu$ can be introduced.
> > >
> > > In section 4, $\widehat{\mu}^{s, a}=\sum_{i=1}^N p_i^\mu 1(x \in \Delta_E)/\Delta$ is an **induced** histogram after we employ the return density decomposition (Eq. 2 in the revised pdf or Eq.3 in the submitted pdf) on a given histogram density function $\widehat{p}^{s, a}$. Note that we use $\ \hat{} \ $ to denote the histogram density estimator to an underlying continuous pdf. In this return density decomposition, given $\epsilon > 1 - p_E$, $x$, and $\widehat{p}^{s, a}$, we can directly evaluate $\widehat{\mu}^{s, a}$ by
> > > $$\widehat{\mu}^{s,a}(x) = \frac{\widehat{p}^{s,a}(x)- (1-\epsilon) 1(x \in \Delta_E)/\Delta}{\epsilon} $$
> > > Importantly, $\widehat{\mu}$ is **the key component of our derived distribution-matching regularization** derived in Proposition 2, which drives the uncertain-aware regularized exploration in applying distributional learning in RL.
> > >
> > >
> > > Overall, we express our gratitude to the reviewer for their consistent dedication to reviewing our work. Should this rebuttal addresses your concerns, we would be grateful for a revised score. Of course, we remain at your disposal for any further clarifications.
> > >
> > > ### Reference
> > >
> > > [1] A Distributional Perspective on Reinforcement Learning (ICML 2017)

---

> > > > ### Author Response · Authors · 2024-11-23
> > > >
> > > > Thank you for your dedication and interest in our paper. As the author and reviewer discussion period approaches its end, we are curious to know your thoughts on our rebuttal and whether you have any additional questions. Should this rebuttal address your concerns, we would be grateful for a revised score. Of course, we remain at your disposal for any further clarifications.

---

> ### Comment · Reviewer_X6Vg · 2024-11-28
>
> Thank you for the detailed response and clarification.
>
> As far as I understand, the induced histogram density function $\mu$ is for the target distribution, and $q$ is the distribution for learning.
> Then, it makes sense that the difference of two distributions represents the uncertainty.
> If it is true, I have more following questions and concerns.
>
> 1. The authors can apply this concept to other distributiononal method, such as QRDQN. Is there any specific reason why the categorical distribution enjoys the proposed concept?
> 2. Bootstrapped DQN can be a important baseline, because it predicts the return distribtuion by bootstrapping multiple heads and shows the deep exploration concept by the uncertainty concept. Can the authors provide some brief explanation on the difference with the proposed method.
> 3. Using the uncertainty of value distribution function for exploration, [A] has already shown that the distribution aware exploration method, PQR outperforms the baseline exploration methods and other distributional RL methods. What is the strength of the proposed method compared to PQR?
> 4. It seems weired that AC (without vanilla entropy) works in mujoco enviornment in Figure 5. It means that the agent performs well without any exploration scheme. Is it correct?
> 5. To consider the manuscript as the deep interpretation of categorial distribution RL, it is natural to adjust the softmax property with temperature to demonstrate why the categorial distribution is benefical than the naive method. However, this paper introduces the additional uncertainty term into the value function for exploration like SAC, which cannot prove the strength of categorical distribution itself.
>
>
> [A] Cho, Taehyun, et al. "Pitfall of optimism: distributional reinforcement learning by randomizing risk criterion." Advances in Neural Information Processing Systems 36 (2024).

---

> > ### Author Response · Authors · 2024-11-30
> >
> > We are grateful to the reviewer for their consistent dedication to reviewing our work. We appreciate your efforts to understand our responses and are pleased that some of the clarifications were useful.
> >
> > It is true that the induced histogram density function $\widehat{\mu}$ is for the target distribution, and $q_\theta$ is the distribution (histogram estimator) for learning. Minimizing the difference between them captures the environmental uncertainty, which is also why we state the decomposed regularization is uncertainty-aware. Here are our further explanations for your concerns.
> >
> >
> > > 1. The authors can apply this concept to other distributiononal method, such as QRDQN. Is there any specific reason why the categorical distribution enjoys the proposed concept?
> >
> > The reason we propose the uncertainty-aware regularized exploration effect and start with categorical distribution is that **this concept is more motivated in CDRL and the CDRL is more theoretically convenient to handle** compared to quantile-based distributional RL.
> >
> > **Stronger Motivation from Literature.** Firstly, there are more clues in the related literature supporting the analysis of the advantage of distributional RL from the perspective of distributional loss. For example, the milestone paper [1] discussed the potential optimization benefits of using KL divergence in distributional RL at the end of their discussion (Section 6). A follow-up work [2] hypothesized that auxiliary tasks could be naturally coupled with reward signals through distributional learning. However, neither of these works conducted an in-depth analysis beyond these discussions. Our paper addresses this gap by introducing the concept of uncertainty-aware regularized exploration effect, directly answering these crucial questions raised in prior literature.
> >
> > **Empirical Similarity with Maximum Entropy RL and Analysis Convenience for KL divergence.** In investigating the benefits of distributional RL, we noticed that MaxEnt RL exhibits similar empirical performance, including faster convergence rates and improved performance in many environments. As we highlighted in Section 5.1, these empirical similarities motivated us to explore their underlying connection, particularly in terms of exploration.
> >
> > Notably, the KL divergence used in CDRL aligns naturally with the entropy maximization principle in MaxEnt RL. This makes CDRL a more appropriate starting point for proposing the uncertainty-aware regularized exploration effect, especially when compared to the vanilla exploration effect in MaxEnt RL. By contrast, quantile-based distributional RL lacks a direct alignment with the maximum entropy framework, which makes the connection to exploration less straightforward. However, there is still potential for future analysis (please refer to Appendix M for our discussion).
> >
> > **Comparison to Quantile-Based Distributional RL.** The benefits of quantile-based distributional RL are more motivated by the robust property of quantile loss against heavy-tailed rewards [3]. Although they also apply quantile TD error decomposition, the proposed technique fundamentally differs from ours, with a divergent explanation of the benefits of using distributional loss in RL. Instead, we provide an extension analysis on quantile-based distributional RL in Appendix M, along with two potential decomposition methods to derive a similar uncertainty-aware regularized exploration to CDRL. Notably, it is more theoretically convenient to handle KL-based CDRL than quantile-based loss, which also explains why we began our concept from CDRL. We leave the detailed extension of our concept to other distributional RL methods, including quantile-based RL, as promising avenues for future research.

---

> ### Author Response · Authors · 2024-11-30
>
> > 2. Bootstrapped DQN can be a important baseline, because it predicts the return distribtuion by bootstrapping multiple heads and shows the deep exploration concept by the uncertainty concept. Can the authors provide some brief explanation on the difference with the proposed method.
>
> We acknowledge that Bootstrapped DQN [4] maintains several independent Q-estimators and randomly samples one of them, enabling the agent to perform temporally extended exploration. The exploration mechanism of Bootstrapped DQN is similar to that of DLTV [6], as both aim to **enhance exploration by leveraging the stochasticity of the return** (as a random variable). Specifically, Bootstrapped DQN uses bootstrapping on the value function in classical RL, while DLTV utilizes the variance from the intrinsically learned return distribution in distributional RL algorithms.
>
> In contrast, our approach decomposes the distributional loss into two parts, with the regularization term being interpreted as a form of exploration that is **intrinsically** promoted when we optimize the distributional loss in distributional RL. This interpretation aligns well the one important exploration strategy in the exploration literature [5], called uncertain-based exploration. The key difference, however, is that while both Bootstrapped DQN and DLTV **explicitly use the ``post-''learned** uncertainty to explore--a relatively common strategy--we argue that distributional RL **intrinsically** promotes uncertainty-aware exploration as a natural consequence of optimizing the distributional loss. This is a novel and interesting connection that bridges the optimization of distributional loss and exploration. Moreover, our explanation offers an intuitive explanation grounded in the exploration literature.
>
>
> > 3. Using the uncertainty of value distribution function for exploration, [A] has already shown that the distribution aware exploration method, PQR outperforms the baseline exploration methods and other distributional RL methods. What is the strength of the proposed method compared to PQR?
>
> We would like to clarify that, similar to DLTV, PQR is also **an improved exploration strategy** by using the estimated uncertainty from distributional RL effectively. As previously explained in our response to Question 2, the purpose of our paper is not to develop advanced exploration based on the learned return distribution from distributional RL. Instead, our decomposed regularization is used to demonstrate that the benefits of using distributional loss in RL are **inherently** promoting an uncertain-aware exploration. Albeit also involving the exploration concept, the primary objective of our paper is fundamentally different from the subsequent exploration strategies based on distributional RL, such as PQR and DLTV.
>
> > 4. It seems weired that AC (without vanilla entropy) works in mujoco enviornment in Figure 5. It means that the agent performs well without any exploration scheme. Is it correct?
>
> For a fair comparison, AC is directly adapted from the implementation of SAC but excludes the entropy term used in SAC. With the remaining architecture, such as the use of target networks, unchanged, this ablation explains why AC still generally performs well. In Figure 5, SAC (AC+VE) outperforms AC in most environments, demonstrating the widely recognized performance improvement (though not in all cases) of incorporating the vanilla entropy term in SAC.
>
> Our implementation is based on RLKit (https://github.com/rail-berkeley/rlkit), a widely recognized reinforcement learning framework in the RL community.

---

> ### Author Response · Authors · 2024-11-30
>
> > 5. To consider the manuscript as the deep interpretation of categorial distribution RL, it is natural to adjust the softmax property with temperature to demonstrate why the categorial distribution is benefical than the naive method. However, this paper introduces the additional uncertainty term into the value function for exploration like SAC, which cannot prove the strength of categorical distribution itself.
>
> We acknowledge that well-known property of the softmax function, which interpolates between a point mass on the value with the **largest output probability** and a concrete uniform distribution, but it is challenging to establish a meaningful connection between this interpolation with the classical RL loss (the naive method in your statement), particularly regarding the largest probability. This contrasts with our proposed return density decomposition, which focuses on the **probability associated with the bin containing the expectation** of the return.
>
> Instead, incorporating the decomposed uncertainty term into the policy gradient framework, as presented in Section 5, is a natural extension immediately following the formed analysis in value-based RL in Section 4. If we do not introduce the additional uncertainty term into the value function, we, alternatively, can also directly analyze the **distributional critic loss** and further apply the return density decomposition again to derive the uncertainty term. However, the subsequent analysis and conclusion after applying this redundant analysis remains the same as the current version, while the current version with a direct comparison to SAC is more readable.
>
>
>
> In summary, thank you again for your review and feedback. We hope our response will help eliminate your concerns. Should this rebuttal address your concerns, we would be grateful for a revised score. Of course, we remain at your disposal for any further clarifications.
>
>
> ### Reference
>
> [1] Bellemare, Marc G., Will Dabney, and Rémi Munos. "A distributional perspective on reinforcement learning." International conference on machine learning. PMLR, 2017.
>
> [2] Lyle, Clare, Marc G. Bellemare, and Pablo Samuel Castro. "A comparative analysis of expected and distributional reinforcement learning." Proceedings of the AAAI Conference on Artificial Intelligence. Vol. 33. No. 01. 2019.
>
>
> [3] Rowland, Mark, et al. "The statistical benefits of quantile temporal-difference learning for value estimation." International Conference on Machine Learning. PMLR, 2023.
>
> [4] Osband, Ian, et al. "Deep exploration via bootstrapped DQN." Advances in neural information processing systems 29 (2016).
>
> [5] Hao, Jianye, et al. "Exploration in deep reinforcement learning: From single-agent to multiagent domain." IEEE Transactions on Neural Networks and Learning Systems (2023).
>
> [6] Mavrin, Borislav, et al. "Distributional reinforcement learning for efficient exploration." International conference on machine learning. PMLR, 2019.

---

### Author Response · Authors · 2024-11-20
**General Author Response**

We thank all reviewers for their in-depth and constructive comments. We have revised and updated our manuscript by following all reviewers' suggestions. We provide the major revision points in the revised paper:

* **Discussion about decomposing quantile-based distributional loss in Appendix M**. Our discussion examines the possibility of extending our decomposition-style analysis to quantile-based distributional loss. This new discussion broadens our research scope and can further enhance our contributions.
* **Enhanced clarity.** We improved the clarity that may not have been accurate or clear in the original submission, including the explanation of $\widehat{\mu}$, comparison with prior work, the equivalence between our work and C51, some steps in the proof, and more details of environments and baselines.
* **More experiments.** We also added more experiments to corroborate our conclusions further, including the ablation study by varying the bin sizes and the equivalence between KL and cross-entropy in our algorithms.

We hope the revised version addresses all the reviewers' concerns. We would appreciate it if the reviewers could check our response to their comments and look through our revised paper further. We are consistently committed to enhancing the quality of our work and remain at your disposal for any further clarifications or questions you might have.

---

### Author Response · Authors · 2024-11-26

Dear Reviewers,

The rebuttal phase is coming to a close. Thank you again for the effort you put into reviewing our paper. We are eager to learn if our rebuttal adequately addresses your concerns. We are happy to provide further clarification and engage in any discussion. Thank you!

Sincerely,
The Authors

---

### Author Response · Authors · 2024-12-01

Dear Reviewers,

The author-reviewer discussion is nearing its final day. We believe we have made our best efforts to address all your concerns, and we remain eager to hear your feedback, even in the final moments of this discussion period. For reviewers to whom we have responded but have not yet had the chance to reply, we sincerely hope our efforts have addressed your concerns and kindly request a revised score if appropriate. Thank you once again for your thorough and thoughtful review of our paper.

Sincerely,
Authors

---

### Meta-Review · Area_Chair_Umq5 · 2024-12-21

**Metareview:**

This paper explores the role of uncertainty-aware exploration in Categorical Distributional RL (CDRL) algorithms, presenting the Distribution-Entropy-Regularized Actor-Critic (DERAC) method. The authors argue that the distributional component serves as a form of regularization, which promotes exploration. The theoretical framework is supported by experimental validation in Atari and MuJoCo environments.

While the paper provides a novel perspective on the benefits of distributional RL, there are still concerns about its clarity, novelty, and scope. Reviewers appreciated the theoretical contributions and the solid experimental validation, but many noted that the presentation could be improved. Some sections, particularly in the theoretical analysis, remain unclear, and there are concerns about vague terminology. Additionally, the focus on DERPI and DERAC, which are not traditional categorical RL algorithms, feels narrow, and there is a need for a broader discussion of more established methods like C51, QR-DQN, and IQN. The novelty of the theoretical contributions, particularly Proposition 3, was also questioned, as it appears similar to existing convergence results in the literature.

The paper would benefit from a clearer explanation of the novelty of the proposed methods, a broader discussion of traditional distributional RL methods, and further clarification of technical terms. Additionally, an ablation study on the impact of bin size would help demonstrate the robustness of the approach.

Overall, the paper has potential but requires revisions to improve its clarity, address novelty concerns, and expand its scope to fully strengthen its contribution to the field.

**Additional Comments On Reviewer Discussion:**

During the rebuttal period, the authors addressed several key concerns raised by the reviewers. Some reviewers appreciated the theoretical contributions and empirical validation but raised issues regarding clarity, novelty, and the scope of the work. The primary concern was that while the paper claims the return distribution contributes to uncertainty-aware exploration, the experiments did not rigorously support this claim. Several reviewers also questioned the novelty of some theoretical results, suggesting they closely resembled prior work. The narrow focus on DERPI and DERAC was another concern, with a suggestion to broaden the comparison to more widely used methods like C51.

The authors responded positively to these concerns, agreeing to improve the clarity of the presentation and expand the scope by comparing their methods with more common distributional RL algorithms. They also promised to conduct an ablation study to address sensitivity to bin sizes and provided further clarification on the theoretical contributions. However, reviewers remained concerned that the paper did not fully demonstrate how the return distribution itself drives uncertainty-aware exploration.

While the authors made valuable responses, further revisions are needed to clarify the contribution, broaden the scope, and provide stronger empirical evidence supporting the claims about the role of return distributions.

---

### Decision · Program_Chairs · 2025-01-22

Reject